# Safeguarding Data in Multimodal AI: A Differentially Private Approach to CLIP Training

## Abstract

The surge in multimodal AI's success has sparked concerns over data privacy in vision-and-language tasks. While CLIP has revolutionized multimodal learning through joint training on images and text, its potential to unintentionally disclose sensitive information necessitates the integration of privacy-preserving mechanisms. We introduce a differentially private adaptation of the Contrastive Language-Image Pretraining (CLIP) model that effectively addresses privacy concerns while retaining accuracy. Our proposed method, DP-CLIP, is rigorously evaluated on benchmark datasets encompassing diverse vision-and-language tasks such as image classification and image captioning. We demonstrate that our approach retains performance on par with the standard non-private CLIP model. Furthermore, we analyze our proposed algorithm under linear representation settings. We derive the convergence rate of our algorithm and show a trade-off between utility and privacy when gradients are clipped per-*batch* and the loss function does not satisfy smoothness conditions assumed in the literature for the analysis of DP-SGD.

## 1 Introduction

The field of vision-language tasks has witnessed a revolutionary breakthrough with the introduction of Contrastive Language–Image Pre-training (CLIP) (Radford et al., 2021) by OpenAI. It has redefined the benchmarks of various downstream vision-language tasks through its exceptional flexibility and remarkable zero-shot learning ability (Li et al., 2021b; Dorbala et al., 2022). CLIP and its successors have been widely used in vision-language tasks such as semantic segmentation, image generation from captions, video summarization, and visual question answering (Galatolo. et al., 2021; Narasimhan et al., 2021; Yao et al., 2021; Li et al., 2021c; Xu et al., 2022; Wang et al., 2022).

However, representations from contrastive learning are rich in information, which might be able to unintentionally memorize more personal sensitive information. Carlini et al. (2023) shows that diffusion models, which are built upon CLIP, can memorize individual images in the training data and emit them at generation time, and is even less private compared to prior generative models such as GANs. Liu et al. (2021) successfully conducts membership inference attacks on the image encoder trained from CLIP. Furthermore, He & Zhang (2021) demonstrates that SimCLR, another framework that employs contrastive learning for image representations, is more susceptible to attribute inference attacks compared to standard supervised models.

Mitigating these privacy vulnerabilities requires privacy-preserving training methods for multimodal models. The field of *differential privacy* (DP) has emerged as the gold standard for privacy-preserving data analytics. It is a mathematical notion of privacy, which ensures that the output distribution of computation is robust to one person's data. DP is typically achieved by introducing randomness in the computation. Differential privacy techniques have been employed in the context of unimodal models (Peng Xu, 2022; Basu et al., 2021; Hoory et al., 2021; Yu et al., 2022; Li et al., 2022b) However, differentially private training of multimodal models is currently understudied. (Carlini et al., 2023) shows directly applying DP-SGD (Abadi et al., 2016) to diffusion models on CIFAR-10 causes the training consistently diverge, even when low privacy guarantees.

In this paper, we propose DP-CLIP, which safeguards vision-language tasks by learning differentially private image and text representations. These representations can then be safely used for various downstream multimodal tasks. It is worth emphasizing that our work represents one of the first attempts to incorporate differential privacy into multimodal models, thus paving the way for enhanced privacy protection in vision-language tasks. We hope that, by leveraging the information contained in the pretrained embeddings using public data, our fine-tuned private representations can maintain their utility in performing various downstream tasks. Since the CLIP loss function involves contrasting data from different pairs, the standard DP deep learning approach, DP-SGD based on *per-sample* clipping, cannot be directly applied. To address this challenge, we employ *per-batch* clipping and show that our algorithm achieves high accuracy while protecting the desired level of privacy, from both theoretical and empirical perspectives.

To demonstrate the strength of our proposed method, we evaluate our differentially private CLIP, DP-CLIP, on benchmark datasets encompassing diverse vision-and-language tasks such as image classification and visual question answering (VQA). The findings demonstrate that our privacy-aware approach retains performance on par with the standard CLIP model while significantly reducing the risk of data exposure.

Furthermore, we theoretically derive the privacy-utility trade-off of DP-CLIP under linear representation settings with linearized loss. Previous works analyzed the convergence rate of DP-SGD under certain smoothness conditions (Yu et al., 2019; Bassily et al., 2019; Feldman et al., 2020; Chen et al., 2020a; Yang et al., 2022; Bu et al., 2022; Fang et al., 2023). However, we note that the CLIP loss function is not convex nor satisfy smoothness conditions in these literature. We also note that we deal with per-batch clipping instead of per-sample clipping analyzed in Abadi et al. (2016); Chen et al. (2020a); Yang et al. (2022); Bu et al. (2022); Fang et al. (2023). Although the loss function does not globally behave well, we exploit the fact that the linearized loss is locally smooth and strongly convex, and provide a probabilistic bound with linear convergence in the number of iterations.

The rest of this paper is organized as follows. We provide the necessary background for interpreting our results in Section 2. In Section 3, we introduce our algorithm DP-CLIP. We present our experiments setup and empirical results in Section 4, followed by our theoretical analysis in Section 5. Finally, we conclude with our discussion in Section 6.

## 2 PRELIMINARIES

**Differential Privacy**    *Differential Privacy* is a mathematical definition of privacy. It is used to enable the analysis of sensitive data while preserving the privacy of individuals, because of its powerful worse-cast guarantees. In essence, differential privacy requires a mechanism's outputs on two adjacent datasets, which differ in one arbitrary person's data, should be indistinguishable. This is achieved by introducing randomness into the computation.

**Definition 1** (Differential Privacy (Dwork et al., 2006))**.** *A randomized algorithm $M : \mathcal{X} \to \mathcal{Y}$ preserves $(\epsilon, \delta)$-differential privacy, if for all adjacent $X, X' \in \mathcal{X}$ s.t. $\forall S \subseteq \mathcal{Y}$,*

$$Pr[M(X) \in S] \leq e^{\epsilon} Pr[M(X') \in S] + \delta. \qquad (2.1)$$

Differentially private mechanisms typically add noise that scales with the *sensitivity* of the function being evaluated. The sensitivity of a function $f$ is defined as the maximum change in $f$ between two neighboring sets: $\Delta f = \max_{X, X' \text{ neighbors}} |f(X) - f(X')|$. The *Gaussian mechanism* with parameters $(\epsilon, \delta)$ takes in a function $q$, dataset $X$, and outputs $f(X) + \mathcal{N}(0, \sigma^2)$, where $\sigma = \sqrt{2 \log(1.25/\delta)} \Delta f / \epsilon$. This canonical mechanism serves as a base for DP-SGD, which is the current state-of-the-art framework for DP deep learning.

Existing works have explored the effects of clipping and noise addition on the convergence and performance of DP deep learning models. Bu et al. (2021) analyzed the impact of per-sample clipping and noise addition on the convergence of DP deep learning models, characterizing these effects through training dynamics and the neural tangent kernel. They also introduced a new technique called *global clipping* that improves the convergence rate of DP-SGD. Wang et al. (2019) studied the convergence properties of DP-SGD and showed that it converges to an approximate local minimum with high probability. Imtiaz & Sarwate (2017) conducted differentially private canonical

correlation analysis (DP-CCA) experiments to evaluate the effectiveness of differential privacy in preventing membership inference attacks. Yu et al. (2019) showed that gradient perturbation is efficient and accurate when the sample size is large and suggested that gradient perturbation may be combined with other differential privacy techniques to achieve even better results.

**CLIP**   The *Contrastive Language-Image Pre-training* (CLIP) (Radford et al., 2021) is a multi-modal vision and language model, trained on a variety of image and text pairs. It is used to produce embeddings for both texts and images. Specifically, let $f : \mathbb{R}^{d_1} \to \mathbb{R}^r$ and $\tilde{f} : \mathbb{R}^{d_2} \to \mathbb{R}^r$ be the dual encoders. Given pairs of data $\{(x_i, \tilde{x}_i)\}_{i \in [n]} \subset \mathbb{R}^{d_1 + d_2}$, the CLIP loss can be formulated as follows:

$$\mathcal{L}(f, \tilde{f}) := -\frac{1}{n} \sum_{i \in [n]} \log \frac{e^{s_{ii}/\tau}}{\sum_{j \in [n]} e^{s_{ij}/\tau}} - \frac{1}{n} \sum_{i \in [n]} \log \frac{e^{s_{ii}/\tau}}{\sum_{j \in [n]} e^{s_{ji}/\tau}}, \tag{2.2}$$

where $s_{ij} := \text{Sim}(f(x_i), \tilde{f}(\tilde{x}_j))$ is the cosine similarity of $x_i$ and $\tilde{x}_j$ measured in the feature space, and $\tau > 0$ is the temperature parameter. The loss in equation 2.2 is a type of *contrastive loss*, that trains encoders by classifying based on whether a pair is observed or artifically paired. Intuitively, the loss 2.2 maps images and texts, that refer to the same object, to vectors with high cosine similarity and map images and texts that are unrelated to vectors with low cosine similarity. We provide the details of the pretaining process to the Appendix B.

## 3   DP-CLIP: PRIVATE AND ACCURATE REPRESENTATIONS

In this section, we introduce our DP-CLIP that incorporates differential privacy into the CLIP model, presented formally in Algorithm 1. Given initial representation functions, the algorithm trains encoders based on per-batch noisy SGD to ensure privacy. The CLIP loss (Radford et al., 2021) is the contrastive cross-entropy loss function computed from pseudo-labels distinguishing whether a pair is observed or artificially generated. Since the CLIP loss function, by definition, contains the similarity between data from multiple pairs, it cannot be written as a sum of losses of individual pairs. Thus we cannot directly apply per-sample clipping technique as in the original DP-SGD. Instead, we employ *per-batch* clipping, and we show that can still guarantee the desired level of privacy, which we discuss in details below.

More specifically, we consider the following setups. Let $f_{\theta_1}$ and $\tilde{f}_{\theta_2}$ be the dual encoders to be trained, parameterized by $\theta_1$ and $\theta_2$, respectively. Let $\theta^{(t)} = (\theta_1^{(t)}, \theta_2^{(t)})$ be the parameters at $t$-th iteration. We obtain a sequence of parameters $(\theta^{(t)})_{t=1}^T$ through mini-batch stochastic gradient descent, where $T$ is the number of iterations. For any subset $\mathcal{B} \subset [n]$, define $\mathcal{L}(\,\cdot\,; \mathcal{B})$ be the loss 2.2 computed only with pairs $\{(x_i, \tilde{x}_i)\}_{i \in \mathcal{B}}$. At iteration $t$, we uniformly sub-sample a mini-batch $\mathcal{B}^{(t)}$ of size $b$ from $[n]$ and compute the partial derivative of $\mathcal{L}(f_{\theta_1}, \tilde{f}_{\theta_2}; \mathcal{B}^{(t)})$, the loss computed with mini-batch, with respect to $\theta$ evaluated at $\theta_1 = \theta_1^{(t)}$ and $\theta_2 = \theta_2^{(t)}$. We denote this as $\partial_\theta \mathcal{L}(f_{\theta_1^{(t)}}, \tilde{f}_{\theta_2^{(t)}}; \mathcal{B})$ for brevity. Then, we clip the mini-batch gradient with a clipping threshold $c > 0$. Let $h^{(t)} = \min\{1, \, c/\|\partial_\theta \mathcal{L}(f_{\theta_1^{(t)}}, \tilde{f}_{\theta_2^{(t)}}; \mathcal{B}^{(t)})\|_F\}$, and we update $\theta^{(t)}$ as follows:

$$\theta^{(t+1)} = \theta^{(t)} - \eta \Big\{ h^{(t)} \partial_\theta \mathcal{L}\big(f_{\theta_1^{(t)}}, \tilde{f}_{\theta_2^{(t)}}; \mathcal{B}^{(t)}\big) + \sigma c \Gamma^{(t)} \Big\},$$

where $\eta > 0$ is the learning rate and $\text{vec}(\Gamma^{(t)}) \sim N(0, I_{rd})$ is the noise added to ensure differential privacy. We formally present the algorithm in 1.

We show that this algorithm is differentially private. In particular, the following result suggests that by carefully choosing $\sigma$, it can achieve desired differential privacy guarantee.

**Proposition 3.1.** *Choose $b < n/10$. There exists universal constants $C_\epsilon, C_\sigma > 0$ such that for any $\epsilon \leq C_\epsilon b^2 T / n^2$ and $\delta > 0$, DP-CLIP is $(\epsilon, \delta)$-differentially private if we choose $\sigma \geq C_\sigma \sqrt{T \log(1/\delta)}/(n\epsilon)$.*

The proof is deferred to the Appendix.

*Remark* 3.1. For implementation of DP-CLIP, we can utilize the existing microbatch SGD (McMahan et al., 2018) implementation in Tensorflow privacy library by setting the number of microbatches

---

**Algorithm 1** DP-CLIP

---

1: **Input:** observed pairs of data $\{(x_i, \tilde{x}_i)\}_{i=1}^n$, number of iterations $T$, noise scale $\sigma$, clipping threshold $c$, learning rate $\eta$, mini-batch size $b$, initial parameters $\theta = (\theta_1^{(0)}, \theta_2^{(0)})$.
2: **for** $t \in \{0, \ldots, T-1\}$ **do**
3:     Uniformly sample mini-batch $\mathcal{B}^{(t)}$ of size $b$ from $[n]$.
4:     **Compute mini-batch gradient**   $g^{(t)} \leftarrow \partial_\theta \mathcal{L}(f_{\theta_1^{(t)}}, \tilde{f}_{\theta_2^{(t)}}; \mathcal{B})$.
5:     **Clip Gradient**   $\bar{g}^{(t)} \leftarrow \min\{1, \ c/\|g^{(t)}\|_F\} g^{(t)}$.
6:     **Add Noise**   $\tilde{g}^{(t)} \leftarrow \bar{g}^{(t)} + \sigma c \mathcal{N}(0, I)$.
7:     **Descent**   $\theta^{(t+1)} \leftarrow \theta^{(t)} - \eta \tilde{g}^{(t)}$.
8: **end for**
9: **return** $\theta^{(T)}$

---

to 1. However, we note that the rationale behind our method and microbatch SGD differs. The microbatch clipping approach is designed to consider the setting of multiple queries per user and enhance training efficiency by leveraging larger number of microbatches. In fact, setting the number of microbatches to 1 in microbatch SGD is atypical (McMahan et al., 2018; Bu et al., 2020; Dupuy et al., 2022). On the other hand, our method specifically implements per-batch clipping due to the fact that the CLIP loss is non-decomposable.

## 4 EXPERIMENTS

In this section, we evaluate our DP-CLIP on image classification and image captioning tasks, and we defer the Visual Question Answering (VQA) task to Appendix C.2. We first introduce the training details in Section 4.1, then we provide a detailed experimental results in Section C.1 and Section 4.3. Our code is available in the supplementary materials.

### 4.1 EXPERIMENTS SETUP

In all of our tasks, given pretrained embeddings, we continue to privately train them using contrastive loss on private datasets (e.g., MNIST, which is treated as private in our paper). We use this two-stage approach to incorporate private data to further update the public pretrained multimodal foundation models. This two-stage approach was also established in Yu et al. (2023), which builds a DP vision foundation model. Apart from their work, we address private multimodal self-supervised learning.

For image classification, we use the pretrained CLIP, and then, for each of the four datasets listed below, we further privately train using DP-CLIP and report the accuracy on the testing set. For each image, we encode the image using CLIP's image encoder and then encode each of the text of the classes with prompt using CLIP's text encoder. We calculate the cosine similarity between the image and text encoders for classes. The predicted class is the one with the highest cosine similarity, following Radford et al. (2021).

We also demonstrate that our DP-CLIP framework can be applied to a broad class of Vision-Language Pre-training models and can therefore be leveraged to perform more complex vision-language tasks. One such task is image captioning. To this end, for image captioning, we use the pretrained *Bootstrapping Language Image Pre-training* (BLIP) Li et al. (2022a), which achieves state-of-the-art performance on a wide range of vision-language tasks, as the backbone. The detailed background of BLIP is deferred to Appendix C. We apply the same per-batch clipping framework and noise injection process described in Algorithm 1 on BLIP loss function. We call this DP-BLIP.

**Datasets** For image classification, we consider four benchmark image classification datasets , namely MNIST (LeCun et al., 1998), Fashion-MNIST (Xiao et al., 2017), CIFAR-10 (Krizhevsky, 2009), and SVHN (Netzer et al., 2011). MNIST is a collection of greyscale images belonging to handwritten digits, with a training set of 60,000 images and a test set of 10,000 images. Fashion-MNIST is a collection of greyscale images of fashion products belonging to 10 categories, with a training set of 60,000 images and a test set of 10,000 images. CIFAR-10 is a collection of color images belonging to one of 10 classes, such as airplane, automobile, and bird, with a training set

of 60,000 images and a test set of 10,000 images. Finally, SVHN is a collection of color images representing street house view numbers, which are images of printed digits of house number plates. We use the training set of 73,257 samples and extra set of 531,131 samples for training, and using testing set of 26,032 samples for evaluation, where the evaluation metric is the classification accuracy. For different datasets, we use different prompts to feed into the model, and we will discuss the details below.

For image captioning task, we employ partial data from the vizwiz image captioning dataset (Gurari et al., 2020), which consists of a training set of 5k image-caption pairs (out of 23k) and a test set of 1k image-caption pairs (out of 8k). We use vizwiz evaluation api to evaluate our results, including BLEU, METEOR, Rouge_L, CIDEr, and SPICE.

**Model Architecture**    For image classification, the base model is OpenAI's CLIP model pretrained on the ImageNet dataset (Deng et al., 2009). It consist of a *ViT-L/14-336px* Transformer architecture as an image encoder and a masked self-attention Transformer as a text encoder. The vision Transformer consists of 24 layers with width of 1024 and 16 heads. The text Transformer consists of 12 layers with width of 768 and 12 heads.

For image captioning, the base model is BLIP pretrained on two human-annotated datasets, COCO (Lin et al., 2014) and Visual Genome (Krishna et al., 2017), and three web datasets, Conceptual Captions (Sharma et al., 2018), Conceptual 12M (Changpinyo et al., 2021), and SBU captions (Ordonez et al., 2011), consisting of an unimodal image encoder, an image-grouded text encoder and an answer decoder, as illustrated in Figure 1b in Appendix C.

**Training Details**    The value of the noise multiplier, $\sigma$, is determined by the size of the training set $n$, the batch size $b$, the number of iterations $T$, and privacy parameters $(\epsilon, \delta)$. Proposition 3.1 provides a guidance on the choice of these parameters. We use TensorFlow Privacy[1] for privacy accounting. Throughout the experiments, we set $\delta = \frac{1}{2n}$ where $n$ is the size of the training set. All our models are implemented in PyTorch (Paszke et al., 2019) using one NVIDIA A100 80G GPU.

## 4.2    IMAGE CLASSIFICATION RESULTS

To obtain optimal results, we conduct hyperparameter tuning and prompt engineering. Because the baseline methods in Papernot et al. (2020) and Vinaroz & Park (2023) did not account for the privacy loss on tuning, we do the same to ensure a fair comparison. We set the learning rate $\eta = 10^{-5}$, since we find it yields the best performance across the datasets. We vary the clipping threshold $c$ from $0.1$ to $1$, and batch size $b$ from 16 to 128. We train the model for 15 to 30 epochs. Regarding prompt engineering, we employ the prompts provided in the CLIP GitHub repository[2]. Our experiments indicated that the inclusion of these prompts enhanced the accuracy of our model by $1 - 2\%$.

Table 1: Evaluation of the Classification Accuracy vs. Privacy of DP-CLIP

|  | MNIST | Fashion-MNIST | CIFAR-10 | SVHN |
|---|---|---|---|---|
| $\epsilon = 10$ | 98.78 | 91.52 | 95.74 | 93.69 |
| $\epsilon = 3$ | 98.56 | 91.27 | 95.62 | 93.23 |
| $\epsilon = 1$ | 98.44 | 90.65 | 94.81 | 92.92 |
| $\epsilon = 0.5$ | 98.40 | 90.02 | 94.47 | 91.03 |
| $\epsilon = 0.25$ | 95.62 | 89.09 | 93.51 | 87.65 |

We present the classification accuracy of DP-CLIP on the four datasets under various $\epsilon$ in table 1. We observe that DP-CLIP is able to recover features under the regime with stringent privacy parameters. In particular, for all datasets, $\epsilon = 1$ performs within $1\%$ of $\epsilon = 10$, which indicates the strong potential of DP-CLIP to offer better privacy guarantees while maintaining utility.

To further evaluate the performance of DP-CLIP, we compare our DP-CLIP against other differentially private image classification methods that achieve state-of-the-art results on these datasets.

---

[1] https://github.com/tensorflow/privacy
[2] https://github.com/openai/CLIP.

We consider DP-Sinkhorn (Cao et al., 2021), DP-KIP (Vinaroz & Park, 2023), DP-SGD with Tempered Sigmoid (DP-SGD (TS) for short) (Papernot et al., 2020), Private-kNN (Zhu et al., 2020), and Active Learning (Zhao et al., 2019), and DP-SGD on over-parameterized models (DP-SGD (large) for short) De et al. (2022). We present the comparisons in Table 2 below. For brevity, only the best results from each paper are included.

Table 2: Comparison with state-of-the-art DP methods on MNIST, FashionMNIST, CIFAR-10 and SVHN, with varying parameter $\epsilon$.

| MNIST | $\epsilon$ | Accuracy | Fashion-MNIST | $\epsilon$ | Accuracy |
|---|---|---|---|---|---|
| DP-Sinkhorn | 10 | 83.2 | DP-Sinkhorn | 10 | 73.0 |
| DP-KIP | 10 | 97.96 | DP-KIP | 10 | 90.2 |
| DP-CLIP | 10 | **98.78** | DP-CLIP | 10 | **91.52** |
| Active Learning | 3 | 97.3 | | | |
| DP-CLIP | 3 | **98.56** | | | |
| DP-SGD (TS) | 2.93 | 98.1 | DP-SGD (TS) | 2.7 | 86.1 |
| DP-KIP | 1 | 97.78 | DP-KIP | 1 | 88.3 |
| DP-CLIP | 1 | **98.44** | DP-CLIP | 1 | **90.65** |
| Private-kNN | 0.47 | **98.8** | | | |
| DP-CLIP | 0.5 | 98.40 | | | |

| CIFAR-10 | $\epsilon$ | Accuracy | SVHN | $\epsilon$ | Accuracy |
|---|---|---|---|---|---|
| DP-SGD (large) | 4 | **96.1** | Active Learning | 6 | 85.0 |
| DP-CLIP | 3 | 95.62 | DP-CLIP | 3 | **91.75** |
| DP-SGD (large) | 1 | 94.7 | Private-kNN | 0.49 | **91.6** |
| DP-CLIP | 1 | **94.81** | DP-CLIP | 0.5 | 91.03 |

From Table 2, we can see that for most cases, DP-CLIP outperforms all other methods on all four datasets while a smaller or equal $\epsilon$. This accuracy improvement is by leveraging both pretraining and using extra caption data for DP-CLIP, which is not present for DP-SGD. Although the performance of Private-kNN is comparable to DP-CLIP on the MNIST and SVHN datasets when $\epsilon < 0.5$. Our DP-CLIP offers more flexibility compared to Private-kNN, as it is only for classification tasks, whereas DP-CLIP can be used for a variety of more complex downstream tasks.

### 4.3  IMAGE CAPTIONING RESULTS

Recent years have seen growing interest in image captioning because of its potential to aid the blind community (Gurari et al., 2020). We adopt BLIP for this task, training the private visual and textual representations jointly using caption-image pairs. Prior to this work, differential privacy has not been applied to image captioning, which is more complex than classification. We establish the first baselines on differentially private image captioning tasks.

We train BLIP privately on a subset of the Vizwiz Image Captioning dataset (Gurari et al., 2020). We present several common evaluation results for various privacy parameters: $\epsilon = 0.0001, 0.5, 5$.

Table 3 compares the image captioning accuracy for different datasets. We include the result from IBM Research AI, which is currently on the top of the VizWiz leaderboard (Code). We can observe that our DP-BLIP achieves comparable performance to the non-private results, and even outperforms them on some metrics. This is likely because our training relies on a model pre-trained on a large public dataset, so the additional private training yields better performance. Another factor is that our private training method preserves representations well and is robust to noise added for privacy. The pre-trained model and robustness of DP-CLIP allow our private approach to achieving strong performance.

Table 3: Evaluation of the Classification Accuracy vs. Privacy of DP-CLIP

|  | $\epsilon = 0.0001$ | $\epsilon = 0.5$ | $\epsilon = 5$ | IBM Research AI (non-private, $\epsilon = \infty$) |
|---|---|---|---|---|
| Bleu_1 | 69.8 | 70.2 | 70.3 | 72.77 |
| Bleu_2 | 51.6 | 52.7 | 54.2 | 54.17 |
| Bleu_3 | 32.5 | 38.3 | 41.4 | 38.97 |
| Bleu_4 | 18.9 | 33.1 | 33.1 | 27.44 |
| ROUGE_L | 43.6 | 45.1 | 45.3 | 50.2 |
| CIDEr | 71.5 | 72.4 | 75.2 | 81.04 |
| SPICE | 10.9 | 12.9 | 13.3 | 17.0 |
| METEOR | 16.9 | 17.9 | 21.4 | 22.25 |

## 5 THEORETICAL ANALYSIS OF DP-CLIP

In this section, we analyze the feature learning capacity of DP-CLIP and derive the privacy-utility trade-off under the linear representation and loss setting. Such a simplified setting has been commonly used in the deep learning theory literature to shed light on understanding complicated deep learning phenomena. For example, the linearized loss function for analyzing representation learning has been used in metric learning (Schroff et al., 2015; He et al., 2018), contrastive learning (Ji et al., 2021) and multimodal contrastive learning (Won et al., 2021; Alsan et al., 2021; Nakada et al., 2023). The linear representation setting has been widely adopted in transfer learning and self-supervised learning (Jing et al., 2021; Tian et al., 2021; Ji et al., 2021; Wu et al., 2022; Tian, 2022; Nakada et al., 2023).

Concretely, suppose that we observe $n$ pairs of data $\{(x_i, \tilde{x}_i)\}_{i=1}^n \subset \mathbb{R}^{d_1} \times \mathbb{R}^{d_2}$. Let $r$ be the dimension of the representation space ($r < d$). We train dual *linear representations* $f(x) = G_1 x$ and $\tilde{f}(x) = G_2 x$, where $G_1 \in \mathbb{R}^{r \times d_1}$ and $G_2 \in \mathbb{R}^{r \times d_2}$, simultaneously with the following contrastive linear loss through noisy gradient descent. For notational brevity, let $G \triangleq [G_1, G_2] \in \mathbb{R}^{r \times d}$, where $d \triangleq d_1 + d_2$.

We aim to obtain $G_1$ and $G_2$ that minimize the following linearized loss function:

$$\mathcal{L}_{\mathrm{L}}(G_1, G_2) = -\frac{1}{n} \sum_i \langle G_1 x_i, G_2 \tilde{x}_i \rangle + \frac{1}{n(n-1)} \sum_{i \neq j} \langle G_1 x_i, G_2 \tilde{x}_j \rangle + \Pi(G), \qquad (5.1)$$

where the penalty term $\Pi(G) \triangleq (\alpha/4)\|GG^\top - I\|_F^2$ with $\alpha > 0$ is added to normalize $G$. Note that the CLIP loss 2.2 becomes equivalent to loss 5.1 without penalty when $\tau \to \infty$.

For observed data $\{(x_i, \tilde{x}_i)\}_{i=1}^n \subset \mathbb{R}^{d_1 + d_2}$, we consider the following spiked covariance model (Johnstone, 2001; Bai & Yao, 2012; Yao et al., 2015; Zhang et al., 2018; Zeng et al., 2019; Ji et al., 2021; Nakada et al., 2023) as the data generation process.

$$x_i = U_1^* z_i + \xi_i, \quad \tilde{x}_i = U_2^* z_i + \tilde{\xi}_i. \qquad (5.2)$$

where $U_1^*$ and $U_2^*$ are $d_1 \times r$ and $d_2 \times r$ orthogonal matrces, respectively. Since the model equation 5.2 is only identifiable up to rotation, we assume that $\Sigma_z$ is a diagonal matrix. Without loss of generality, we further assume that $\|\Sigma_z\| = 1$. We assume that $z_i$, $\xi_i$ and $\tilde{\xi}_i$ are mean 0 sub-Gaussian random variables with parameters bounded by a universal constant. Furthermore, we assume the independence of variables; $z_i \perp\!\!\!\perp \xi_i$, $\tilde{z}_i \perp\!\!\!\perp \tilde{\xi}_i$, and $\xi_i \perp\!\!\!\perp \tilde{\xi}_i$.

There have been several works on analyzing the convergence of DP-SGD (Abadi et al., 2016) with per-sample clipping (Yu et al., 2019; Bassily et al., 2019; Feldman et al., 2020; Chen et al., 2020a; Yang et al., 2022; Bu et al., 2022; Fang et al., 2023). Closely related works are Yang et al. (2022) and Fang et al. (2023). However, the contrastive loss function of CLIP cannot be decomposed into the sum of per-sample losses, since CLIP learns by contrasting modalities across sampled pairs. This is the reason why DP-CLIP employs per-*mini-batch* clipping. In addition, in spite of the linearization, our loss function equation 5.1, like the original CLIP loss, is neither convex nor globally Lipschitz, which makes the theoretical analysis highly nontrivial.

## 5.1 PRIVACY-UTILITY TRADE-OFF OF DP-CLIP

Let $G_1^*, G_2^*$ be the minimizer of the population loss $\mathbb{E}[\mathcal{L}_L(G)]$. For simplicity, we assume the regularization parameter $\alpha = \Theta(1)$ is of constant order. Since pretrained encoder is often used in downstream tasks, where the output of the encoder is fed into neural networks or linear probes, the essential information of the learned representation is contained in the linear transformation. For this reason, we measure the performance of the learned representations through the excess loss of information defined as $\min_{A \in \mathbb{R}^{r \times r}} \|AG_1 - G_1^*\|_F \vee \min_{A \in \mathbb{R}^{r \times r}} \|AG_2 - G_2^*\|_F$. For "good" encoders, we expect that a certain linear transformation of it is close to the representations obtained using infinite number of training samples.

Before presenting our results, we introduce notations. For two sequences of positive numbers $\{a_k\}_k$ and $\{b_k\}_k$, we write $a_k \lesssim b_k$ if and only if there exists a constant $C > 0$, independent of the index $k$, such that $\sup_{k \in \mathcal{K}}(a_k/b_k) < C$. Moreover, we write $a_k \ll b_k$ when $\sup_{k \in \mathcal{K}}(a_k/b_k) \leq C_u$ holds for a sufficiently large universal constant $C_u > 0$ common throughout the paper. For any matrix $A$, we denote $\|A\|$ and $\|A\|_F$ as the operator norm and Frobenius norm of $A$ respectively. For any matrix $A$, let $\lambda_{\min}(A)$ and $\lambda_{\max}(A)$ be the minimum and maximum singular values of $A$, respectively. For any zero-mean random variable $X$, we define its covariance matrix as $\Sigma_X \triangleq \mathbb{E}[XX^\top]$. Let the signal-to-noise ratio for $x$ and $\tilde{x}$ be $s_1^2 \triangleq \|\Sigma_z\|/\|\Sigma_\xi\|$ and $s_2^2 \triangleq \|\Sigma_z\|/\|\Sigma_{\tilde\xi}\|$, respectively.

*Assumption* 5.1. Assume that $d > r$ and $n \gg r(r + s_1^{-2} r_e(\Sigma_\xi) + s_2^{-2} r_e(\Sigma_{\tilde\xi}))^2 \log^3(T(n + d))$, where $r_e$ is the effective rank defined as $r_e(A) \triangleq \mathrm{Tr}(A)/\|A\|$ for any square matrix $A$.

*Assumption* 5.2 (Signal-to-noise Ratio). Assume that $\min\{s_1^2, s_2^2\} \gtrsim 1$.

*Assumption* 5.3 (Signal Condition Number). Assume that $\kappa \triangleq \lambda_{\max}(\Sigma_z)/\lambda_{\min}(\Sigma_z) \lesssim 1$.

Assumption 5.1 ensures that we have an effective number of samples to separate the core signal from the noise. Assumption 5.2 is a mild condition on the signal-to-noise ratio. It allows the noise to be the same strength as signal. Assumption 5.3 ensures that core features are strongly shared between the two modalities.

Here we introduce the privacy-utility trade-off of DP-CLIP under linear loss.

**Theorem 5.1** (Privacy-utility Trade-off). *Suppose Assumptions 5.1, 5.2 and 5.3 hold. Assume that $\alpha = \Theta(1)$. Let $G^{(T)}$ be the representation obtained from the algorithm 1 with loss $\mathcal{L}_L(G)$. Suppose that the initial representation $G^{(0)}$ satisfies $\min_{O \in \mathbb{R}^{r \times r}: O^\top O = I} \|OG^{(0)} - \hat{G}\|_F \ll 1$. Choose $c \gg 1$, $b = \lceil \nu n \rceil$, where $\nu \in (0, 1)$ is a constant. Also choose $\eta > 0$ and $\sigma > 0$ as $\eta = \{\sigma \sqrt{T(rd + \log(T(n + d)))}\}^{-1}, \sigma = C_\sigma \sqrt{T \log(1/\delta)}/(n\epsilon)$, where $C_\sigma$ is a constant appearing in Proposition 3.1. Then, Algorithm 1 under loss $\mathcal{L}_L$ is $(\epsilon, \delta)$-DP and for sufficiently large $T$,*

$$\min_{A \in \mathbb{R}^{r \times r}} \|AG_1^{(T)} - G_1^*\|_F \vee \min_{A \in \mathbb{R}^{r \times r}} \|AG_2^{(T)} - G_2^*\|_F$$

$$\lesssim \underbrace{\exp\left(-\frac{n\epsilon}{8\kappa C_\sigma \sqrt{\log(1/\delta)\{rd + \log(T(n + d))\}}}\right)}_{\text{optimization error}} + \underbrace{\frac{\log^{1/4}(1/\delta)\{rd + \log(T(n + d))\}^{1/4}}{\sqrt{n\epsilon}}}_{\text{cost of privacy}}$$

$$+ \underbrace{\sqrt{\frac{r(r + s_1^{-2} r_e(\Sigma_\xi) + s_2^{-2} r_e(\Sigma_{\tilde\xi}))^2 \log^3(n + d)}{n}}}_{\text{statistical error}}. \tag{5.3}$$

*holds with probability at least $1 - O((n + d)^{-1})$.*

**Proof Outline of Theorem 5.1.** This result follows from the linear convergence result (Theorem D.1) of Algorithm 1; we can bound the distance between $G^{(T)}$ and the global minimizer of the loss $\mathcal{L}_L$ by three components, a linear converging term, the error from the injected noise, and the error from subsampling. To show this result, we first derive the one-step linear convergence bound for non-stochastic gradient descent without noise injection. To this end, we use the fact that $\mathcal{L}_L$ is locally strongly convex and directionally smooth around its global minimum. For noisy stochastic gradient descent, we need to control the accumulation of errors coming from both privacy noise and subsampling. For this purpose, we exploit the fact that $\partial_G \mathcal{L}_L(G^{(t)}; \mathcal{B}^{(t)})$, a mini-batch gradient of $\mathcal{L}_L$ evaluated at $G^{(t)}$, is an unbiased estimator of $\partial_G \mathcal{L}_L(G^{(t)})$, and control the deviation of the

accumulated errors through the martingale concentration inequality. The accumulated error from subsampling is controlled by the Bernstein concentration bound from Bardenet & Maillard (2015), which turns out to be negligible since the batch size is chosen to be proportional to the number of samples. Given the linear convergence result, we set the value of $\eta$ and $\sigma$ as specified in Proposition 3.1 to conclude the proof. The proof and more detailed statement of Theorem 5.1 that specifies the exact condition on $T$ is available in Corollary D.2 in the appendix.

In equation 5.3, the right-hand side consists of three terms: optimization error, privacy cost, and statistical error. The optimization error decreases exponentially in $n$, since the loss function $\mathcal{L}_{\mathrm{L}}$ behaves well locally around the global minimum. This term grows with $T$ because the algorithm runs in two stages: in the first stage, the error decreases exponentially in $T$ (see details in Theorem D.1), and in the second stage, when $G^{(T)}$ reaches a certain stable region, the optimization error starts to increase (slowly) if we continue to run more gradient descent updates. We also note that the optimization error is dominated by the term for the cost of privacy when $n\epsilon/\sqrt{\log(1/\delta)}$ is large. The second term corresponds to the additional cost to preserve privacy. Ignoring the logarithmic term $\log(T(n+d))$, the cost increases proportionally to $\log^{1/4}(1/\delta)/\epsilon^{1/2}$. This rate also appears similarly in Chen et al. (2020a); Yang et al. (2022); Fang et al. (2023) for the analysis of DP-SGD. Our technical analysis differs from theirs as they consider per-sample clipping for loss functions that satisfy certain smoothness condition, which does not hold for our loss 5.1. We also note that the loss 5.1 is not decomposable to apply per-sample clipping. Also, similar privacy cost bounds appear in the convergence analysis of differentially private gradient descent and related algorithms (Wang et al., 2017; 2019; Zhang et al., 2019; Wang et al., 2020; Cai et al., 2021; 2023). The statistical error term is due to the irreducible error coming from finite samples. The term depends on $r + s_1^{-2}r_e(\Sigma_\xi) + s_2^{-2}r_e(\Sigma_{\tilde\xi})$, which is trivially bounded by $O(r + d)$ under Assumption 5.2. When either the effective rank of noise covariance is small or the signal-to-noise ratio is large, the statistical error term becomes small.

*Remark* 5.1. In the analysis, the intrinsic dimension $r$ of the input data is assumed to be known. In practical situation, we can either choose $r$ based on certain metric such as cross validation in downstream tasks, or estimate $r$ based on the spectral decay of the cross-covariance matrix.

*Remark* 5.2. The initial value condition is satisfied if $\min_{O:O^\top O=I}\|OG^{(0)} - \hat{G}\| \ll 1/\sqrt{r}$. Thus the condition is considered to be weak when $r$ is small. As in our experiments in Section 4, we can employ initial representations trained with non-private optimizers with certain number of samples.

## 6 DISCUSSION

In this paper, we introduce DP-CLIP, a novel approach that integrates differential privacy into the CLIP to address privacy concerns associated with vision-language tasks. To our knowledge, this is the first attempt to apply differential privacy approaches to multimodal training, where we have created a framework applicable to a variety of vision-image tasks that have not previously been explored in the DP literature. We conduct extensive experiments to demonstrate the effectiveness of our DP-CLIP on three tasks: image classification, image captioning and visual question answering tasks. In addition, we theoretically prove the convergence of our algorithm under linear representation settings, and present a privacy-utility trade-off for DP-CLIP in the situation where gradients are clipped per batch and the loss function does not satisfy the smoothness conditions such as Lipschitz smoothness.

Our work identifies several areas for further investigation. Future work includes conducting experiments on privatizing other multimodal models and evaluating them across a broader range of vision-language downstream tasks, such as visual entailment or image-text retrieval (Li et al., 2021a). Although previous work has suggested that DP can protect against common privacy attacks such as reconstruction attacks and membership inference attacks (Chen J, 2021), investigating the empirical privacy auditing on DP-CLIP is also interesting. Additionally, while our theoretical analysis focuses on linear representation functions, non-linear representations need further exploration. Moreover, Vyas et al. (2023) introduced a copyright protection framework for generative models, which relates to differential privacy as a mathematical measurement. Given that our method learns differentially private image representations, it has the potential to generate copyright-protected images.

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

## A    APPENDIX

## B    ADDITIONAL PRELIMINARIES ON CLIP

The goal of CLIP is to train an image encoder and a text encoder, by maximizing the cosine similarity of correct image-text pairs (highlighted entries on the diagonal in Fig. 1a) and minimizing the cosine similarity of incorrect image-text pairs (other non-diagonal entries in Fig. 1a.)

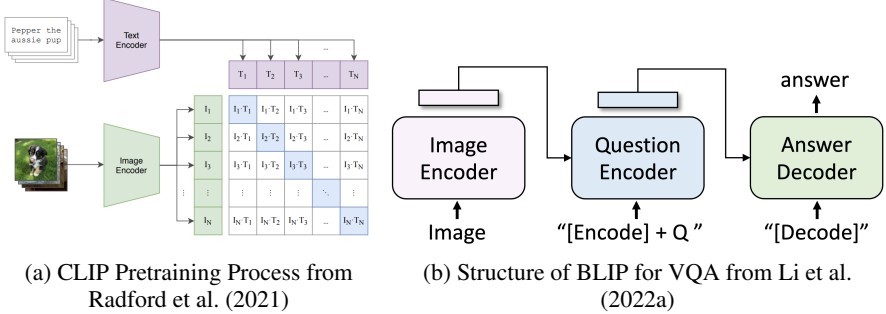

| (a) CLIP Pretraining Process from Radford et al. (2021) | (b) Structure of BLIP for VQA from Li et al. (2022a) |

## C    DETAILED EXPERIMENTAL RESULTS

In this section, we provide additional experimental setups and results.

### C.1    IMAGE CLASSIFICATION

We present the hyperparameters used in image classification task in Table 4.

Table 4: Tuned Hyperparameters for Image Classification DP-CLIP

|              | MNIST       | Fashion-MNIST | CIFAR-10    | SVHN        |
|--------------|-------------|---------------|-------------|-------------|
| lr           | 1e-05       | 1e-05         | 1e-05       | 1e-05       |
| betas        | (0.9, 0.98) | (0.9, 0.98)   | (0.9, 0.98) | (0.9, 0.98) |
| eps          | 1e-06       | 1e-06         | 1e-06       | 1e-06       |
| weight decay | 0.01        | 1e-06         | 1e-06       | 1e-06       |
| num epochs   | 30          | 30            | 30          | 15          |
| batch size   | 32          | 32            | 32          | 32          |

### C.2    VQA

The *Bootstrapping Language Image Pre-training* (BLIP) Li et al. (2022a) is a novel framework for VLP (Vision-Language Pre-training Gan et al. (2022)) that offers broad applicability to various downstream tasks. It introduces (a) *Captioning and Filtering* (CapFilt), a pioneering method for dataset bootstrapping that enables learning from noisy image-text pairs, and (b) *Multimodal mixture of Encoder-Decoder* (MED), which is a novel model architecture capable of functioning as a unimodal encoder, an image-grounded text encoder, or an image-grounded text decoder. This versatility facilitates effective multi-task pre-training and flexible transfer learning. The MED model is jointly pretrained using three vision-language objectives: *image-text contrastive learning* (Radford et al., 2021) to align the vision and language representations, *image-text matching* (Li et al., 2019) to distinguish between positive and negative image-text pairs, and *image-conditioned language modeling*(Kiros et al., 2014) to generate good textual descriptions given an image. We decided to use BLIP as our backbone since it currently achieves state-of-the-art performance on the image captioning task.

Visual Question Answering (VQA) has been increasingly used in many fields, such as healthcare, education, and social media (Srivastava et al., 2021). In Visual Question Answering (VQA), an

image (I) is provided along with a related question (Q) in natural language form, and the goal is to generate accurate and meaningful answers (A). However, the images and text data used in training may contain sensitive information, and it is of paramount importance to ensure privacy preservation measures are in place (Bara et al., 2022). For similar reasons as the image-captioning section in 4.2, because BLIP shows such strong performance on VQA tasks, we decided to use it as the backbone for our VQA experiments.

For VQA tasks, instead of framing it as a multi-answer classification problem Chen et al. (2020b); Li et al. (2020), BLIP takes a different approach by formulating it as an answer generation task (Li et al., 2021a; 2022a). This formulation allows for open-ended VQA, where the model generates answers rather than selecting from a predefined set of options, which is consistent with the task used in Li et al. (2022a) and Li et al. (2021a). As depicted in Figure 1b, during the finetuning process, an image-question pair is encoded into multimodal embeddings, which are then fed into an answer decoder. We use contrastive loss on the question-answer and image pairs to train private representation and then without further finetuning, we evaluate our results using the exact-match accuracy metric.

The goal of the experiment in this section is to demonstrate that adding DP noise does not significantly impact the accuracy of the results, compared to the non-private method, when all other parameters are held constant. The objective is not to fine-tune the model to compete with non-private state-of-the-art results, but rather to showcase that our approach achieves a comparable level of accuracy without compromising privacy. We note that BLIP can achieve state-of-the-art results on VQA with a much higher accuracy rate of $78.25\%$ (Li et al., 2022a). However, in our study, we deliberately refrain from extensive parameter optimization and instead focus on providing a baseline analysis. As a result, we report lower accuracy results compared to the fine-tuned BLIP approach. From Table 5, we can see that our model maintains utility even as privacy measures are increased, suggesting its resilience to noise.

Table 5: Evaluation of VQA on the abstract scene VQA2.0 dataset of DP-BLIP.

| $\epsilon = \infty$ (non-private) | $\epsilon = 10$ | $\epsilon = 3$ | $\epsilon = 1$ | $\epsilon = 0.5$ |
|---|---|---|---|---|
| 55.22% | 52.95% | 52.94% | 52.94% | 52.94% |

We report the exact-match accuracy mentioned in C.2, here, we additionally evaluate it using another metric that is robust to iter-human variability in phrasing the answers, introduced in `https://visualqa.org/evaluation.html`. The evaluation code was taken from ALBEF's github repository, where we consider the top three answers given by humans to a question and our accuracy is taken to be $\min\{1, \#\text{ humans saying that answer}/3\}$ and we output the average accuracy over the training set. The following results, shown in Tab. 6, suggests our privacy-aware approach can achieve comparable performance to the non-private method.

Table 6: Evaluation of VQA Accuracy vs. Privacy of DP-BLIP with Top Three Answers

| $\epsilon = 0.0001$ | $\epsilon = 0.5$ | $\epsilon = 5$ | $\epsilon = 50$ |
|---|---|---|---|
| 0.4686 | 0.4717 | 0.4784 | 0.4924 |

# D    PROOFS OF THEORETICAL RESULTS

Before going into the proofs, we introduce notations to be used in later sections.

## D.1    NOTATION

In this section, we introduce notations to be used. We write $a_k = O(b_k)$ if $a_k \lesssim b_k$ holds and $a_k = \Omega(b_k)$ if $a_k \gtrsim b_k$ holds. $\mathbb{O}_{d,r} \triangleq \{O \in \mathbb{R}^{r \times d} : O^\top O = I_r\}$ as a set of orthogonal matrices of order $d \times r$. We write $a \vee b$ and $a \wedge b$ to denote $\max(a, b)$ and $\min(a, b)$, respectively. For any matrix $A$, let $\lambda_j(A)$ be the $j$-th largest singular value of $A$. Let $\lambda_{\min}(A)$ and $\lambda_{\max}(A)$ be the minimum

and maximum singular values of $A$, respectively. Moreover, for any square matrix $A$, define its effective rank as $r_e(A) = \text{Tr}(A)/\|A\|$. For any zero-mean random variables $X$ and $\tilde{X}$, we define the covariance matrix of $X$ as $\Sigma_X \triangleq \mathbb{E}[XX^\top]$, and the cross-covariance matrix of $X$ and $\tilde{X}$ as $\Sigma_{X,\tilde{X}} \triangleq \mathbb{E}[X\tilde{X}^\top]$. Define $\hat{\Sigma}_{x,\tilde{x}}$ as $\hat{\Sigma}_{x,\tilde{x}} \triangleq 1/n \sum_{i\in[n]} x_i\tilde{x}_i^\top - 1/n/(n-1) \sum_{i\neq j} x_i\tilde{x}_j^\top$.

Here we prove results in 5. We consider minimizing the following linear loss function:

$$\mathcal{L}_{\text{L}}(G) \triangleq - \text{tr}\Big(G_1\hat{\Sigma}_{x,\tilde{x}}G_2^\top\Big) + \Pi(G),$$

where $\Pi(G) = (\alpha/4)\|GG^\top - I_r\|_F^2$ with $\alpha > 0$. We also define the loss for mini-batch $\mathcal{B}$ as $\mathcal{L}_{\text{L}}(G;\mathcal{B}) \triangleq - \text{tr}\Big(G_1\hat{\Sigma}_{x,\tilde{x},\mathcal{B}}G_2^\top\Big) + \Pi(G)$, where

$$\hat{\Sigma}_{x,\tilde{x},\mathcal{B}} \triangleq \frac{1}{|\mathcal{B}|} \sum_{i\in\mathcal{B}} x_i\tilde{x}_i^\top - \frac{1}{|\mathcal{B}|(|\mathcal{B}|-1)} \sum_{i\neq j; i,j\in\mathcal{B}} x_i\tilde{x}_j^\top.$$

## D.2 Differential Privacy of DP-CLIP

Here we present the Gaussian mechanism, the theoretical foundation of DP-SGD and DP-Adam above.

**Definition 2** (Gaussian Mechanism). *(Dwork et al., 2014) Let $f : \mathcal{X} \to \mathbb{R}^d$ be an arbitrary d-dimensional function, i.e. $f(x) = [f_1(x), f_2(x), ..., f_d(x)]$ for $x \in \mathcal{X}$. Then, the Gaussian mechanism with parameter $\sigma$ outputs,*

$$M(x) = [f_1(x) + Z_1, f_2(x) + Z_2, ..., f_d(x) + Z_d],$$

*where $Z_i \sim \mathcal{N}(0, \sigma^2)$ for $i \in [d]$.*

**Definition 3** ($\ell_2$-sensitivity). *The $\ell_2$-sensitivity of a function $f : \mathcal{X} \to \mathcal{Y}$ is defined as,*

$$\Delta_2(f) = \max_{adjacent\ x_1,x_2\in\mathcal{X}} \|f(x_1) - f(x_2)\|_2.$$

We then present the privacy guarantee for DP-CLIP.

**Proposition 4.** *Choose for $b < n/10$. There exists universal constants $C_\epsilon, C_\sigma > 0$ such that for any $\epsilon \leq C_\epsilon b^2 T/n^2$ and $\delta > 0$, DP-CLIP is $(\epsilon,\delta)$-differentially private if we choose $\sigma \geq C_\sigma\sqrt{T\log(1/\delta)}/(n\epsilon)$.*

*Proof.* Note that at each iteration, we can view Algorithm 1 as a repeated composition of subsampling and Gaussian mechanism. Let $\mathcal{M} = \mathcal{M}_T \circ \mathcal{M}_{T-1} \circ \cdots \circ \mathcal{M}_1$, where $\mathcal{M}_t \triangleq \mathcal{M}_{t,G} \circ \mathcal{M}_{t,s}$ is the composition of subsampling and Gaussian mechanism at $t$-th iteration. We first bound the $\ell_2$ sensitivity for Gaussian mechanism $\mathcal{M}_{t,G}$. Note that $\mathcal{M}_{t,G}$ depends on the mini-batch $\mathcal{B}^{(t)} \subset [n]$. We write $\bar{g}^{(t)} = \bar{g}^{(t)}((x_i, \tilde{x}_i)_{i\in\mathcal{B}^{(t)}})$ to make explicit the dependence of $g^{(t)}$ on the pairs of data $(x_i, \tilde{x}_i)_{i\in\mathcal{B}^{(t)}}$. Note that we can bound the $\ell_2$ sensitivity as

$$\max_{i\in\mathcal{B}^{(t)}} \sup_{(x_i,\tilde{x}_i),(x_i',\tilde{x}_i')\in\mathbb{R}^{d_1+d_2}} |\bar{g}^{(t)}(\ldots,(x_i,\tilde{x}_i),\ldots) - \bar{g}^{(t)}(\ldots,(x_i',\tilde{x}_i'),\ldots)| \leq 2c$$

due to per-batch clipping. The rest of the proof follows from the result of privacy amplification by subsampling (Theorem 11 from Bun et al. (2018)), and a similar argument in the proof of Lemma 3.1 from Yang et al. (2022). $\square$

## D.3 Optimization Error Bound

In this subsection, we aim to derive the optimization error bound for $\text{dist}(G^{(T)}, \hat{G})$ given fixed pairs of data $\{(x_i, \tilde{x}_i)\}_{i=1}^n$, where the distance $\text{dist}$ is defined as follows: For any matrices $A \in \mathbb{R}^{r\times d}$ and $A' \in \mathbb{R}^{r\times d}$, define the distance as

$$\text{dist}(A, A') \triangleq \min_{O\in\mathbb{O}_{r,r}} \|OA - A'\|_F.$$

*Assumption* D.1 (Local Directional Strong Convexity of $\mathcal{L}_L$). Assume that there exists some $\gamma > 0$ such that for any $G$ satisfying $\|G - \hat{G}\|_F \leq \gamma$, the following inequalities hold for all $Z \in \mathbb{R}^{r \times d}$:

$$\text{vec}(Z)^\top \frac{\partial^2 \mathcal{L}_L(G)}{\partial \text{vec}(G) \partial \text{vec}(G)^\top} \text{vec}(Z) \leq \beta_u \|Z\|_F^2. \tag{D.1}$$

$$\text{vec}(H_Z Z - \hat{G})^\top \frac{\partial^2 \mathcal{L}_L(G)}{\partial \text{vec}(G) \partial \text{vec}(G)^\top} \text{vec}(H_Z Z - \hat{G}) \geq \beta_l \|HZ - \hat{G}\|_F^2, \tag{D.2}$$

where $H_Z \triangleq \arg\min_{O \in \mathbb{O}_{r,r}} \|OZ - \hat{G}\|_F$.

Before presenting the theorem and its proof, we list lemmas to be used in the proof. The proofs of lemmas are deferred to Section E.

**Lemma D.1.** *Suppose that Assumption D.1 holds with triple $(\beta_u, \beta_l, \gamma)$ and that*

$$\text{dist}^2(G, \hat{G}) \leq \gamma^2. \tag{D.3}$$

*Let $H \triangleq \arg\min_{H \in \mathbb{O}_{r,r}} \|HG - \hat{G}\|_F$. Define $\bar{G} \triangleq G - \eta \partial_G \mathcal{L}_L(G)$ and $\tilde{G} \triangleq G - \eta g$, where $g \in \mathbb{R}^{r \times d}$ is any matrix. If $\eta \leq 1/\beta_u$, then,*

$$\|H\bar{G} - \hat{G}\|_F^2 \leq (1 - \eta\beta_l) \text{dist}^2(G, \hat{G}),$$
$$\|H\tilde{G} - \hat{G}\|_F^2 \leq (1 - \eta\beta_l) \text{dist}^2(G, \hat{G}) + 2\eta\langle H\bar{G} - \hat{G}, \partial_G \mathcal{L}_L(G) - g \rangle + \eta^2 \|g - \partial_G \mathcal{L}_L(G)\|_F^2.$$

**Lemma D.2.** *Let $\mathcal{B} \subset [n]$ be a uniformly sampled random batch of size $b$ in $[n]$. Then,*

$$\mathbb{E}_{\mathcal{B}}[\partial_G \mathcal{L}_L(G; \mathcal{B})] = \partial_G \mathcal{L}_L(G)$$

*holds for all $G \in \mathbb{R}^{r \times d}$, where the expectation is taken with respect to subsampling.*

Define $R \triangleq (\max_{i \in [n]} \|x_i\|)(\max_{i \in [n]} \|\tilde{x}_i\|)$.

**Lemma D.3.** *Fix $T > 0$. Suppose that $\max_{t \in [T]} \text{dist}(G^{(t)}, \hat{G})^2 \leq \gamma^2$ holds. Then,*

$$\max_{t \in [T]} \|\partial_G \mathcal{L}_L(G^{(t)}; \mathcal{B}^{(t)}) - \partial_G \mathcal{L}_L(G^{(t)})\|_F \lesssim (\|\hat{G}\|_F + \gamma) R \left( \sqrt{\frac{(1 - b/n) \log(T(n+d))}{b}} + \frac{1}{b} \right)$$

*holds with probability $1 - O((n+d)^{-1})$.*

**Lemma D.4.** *Suppose that $x_i, \tilde{x}_i$ are generated according to the model in equation 5.2. Suppose that $\max_{t \in [T]} \text{dist}^2(G^{(t)}, \hat{G}) \leq \gamma^2$, where $\gamma$ satisfies $\gamma \leq 1 \wedge 1/\alpha$. Then,*

$$\max_{t \in [T]} \|\partial_G \mathcal{L}_L(G^{(t)}; \mathcal{B}^{(t)})\|_F \lesssim (\sqrt{r}\|\hat{G}\| + 1) R \sqrt{\frac{\log(T(n+d))}{b}} + \gamma\|\hat{\Sigma}_{x,\tilde{x}}\| + \alpha(\|\hat{G}\|^2 + 1)\gamma$$

*holds with probability $1 - O((n+d)^{-1})$.*

Using above lemmas, we obtain the following theorem.

**Theorem D.1.** *Suppose that Assumption D.1 holds with triple $(\beta_u, \beta_l, \gamma)$ and that*

$$\text{dist}^2(G^{(0)}, \hat{G}) \leq \frac{\gamma^2}{8}. \tag{D.4}$$

*We obtain a sequence of representations $(G^{(t)})_{t \in [T]}$ from noisy mini-batch SGD according to Algorithm 1 with linear loss $\mathcal{L}_L$. Set the clipping threshold $c$ and the mini-batch size $b$ as*

$$c \gg (\sqrt{r}\|\hat{G}\| + 1) R \sqrt{\frac{\log(T(n+d))}{b}} + \gamma\|\hat{\Sigma}_{x,\tilde{x}}\| + \alpha(\|\hat{G}\|^2 + 1)\gamma, \tag{D.5}$$

$$b \gg \frac{1}{\gamma^2} (\sqrt{r}\|\hat{G}\| + \gamma)^2 R^2 \log(T(n+d)). \tag{D.6}$$

*If $\eta > 0$ satisfies*

$$\eta \le \min\left\{\frac{1}{2\beta_u}, \ \frac{\beta_l \gamma^2}{4\sigma^2 c^2 (2rd + 60\log(T(n+d)))}\right\}, \tag{D.7}$$

*then,*

$$\text{dist}^2(G^{(T)}, \hat{G}) \lesssim (1 - \eta\beta_l)^T \text{dist}^2(G^{(0)}, \hat{G}) + \frac{\eta\sigma^2 c^2}{\beta_l}(rd + \log(T(n+d)))$$

$$+ \frac{\eta}{\beta_l}(\sqrt{r}\|\hat{G}\| + \gamma)^2 R^2 \left(\sqrt{\frac{(1 - b/n)\log(T(n+d))}{b}} + \frac{1}{b}\right)^2.$$

*holds with probability $1 - O((n+d)^{-1})$.*

*Proof of Theorem D.1.* For notational brevity, write $g^{(t)} = \partial_G \mathcal{L}_{\text{L}}(G^{(t)}; \mathcal{B}^{(t)})$. Define $H^{(t)} \triangleq \arg\min_{H \in \mathbb{O}_{r,r}} \|HG^{(t)} - \hat{G}\|_F$. Also define $\tilde{G}^{(t+1)} := G^{(t)} - \eta g^{(t)}$. From equation D.5 and Lemma D.4,

$$\max_{t \in [T]} \|\partial_G \mathcal{L}_{\text{L}}(G^{(t)}; \mathcal{B}^{(t)})\|_F \le c$$

holds with probability $1 - O((n+d)^{-1})$. Henceforth, we focus on this event, where $h^{(t)} = 1$ holds for all $t \in [T]$. Observe that

$$\text{dist}^2(G^{(t+1)}, \hat{G}) \le \|H^{(t)} G^{(t+1)} - \hat{G}\|_F^2$$

$$= \|H^{(t)}\tilde{G}^{(t+1)} - \hat{G} - \eta\sigma c H^{(t)}\Gamma^{(t)}\|_F^2$$

$$= \|H^{(t)}\tilde{G}^{(t+1)} - \hat{G}\|_F^2 - 2\eta\sigma c \,\text{tr}\left((H^{(t)}\tilde{G}^{(t+1)} - \hat{G})^\top H^{(t)}\Gamma^{(t)}\right) + \eta^2\sigma^2 c^2 \|\Gamma^{(t)}\|_F^2. \tag{D.8}$$

Define $D^{(t+1)} \triangleq \|H^{(t)}\tilde{G}^{(t+1)} - \hat{G}\|_F$. Observe that

$$-2\eta\sigma c \,\text{tr}\left((H^{(t)}\tilde{G}^{(t+1)} - \hat{G})^\top H^{(t)}\Gamma^{(t)}\right) = -2\eta\sigma c \sum_{j \in [r], k \in [d_1]} (H^{(t)\top}(H^{(t)}\tilde{G}^{(t+1)} - \hat{G}))_{jk}(\Gamma^{(t)})_{jk}$$

$$\triangleq 2\eta\sigma c D^{(t+1)} u^{(t)}.$$

Also,

$$\eta^2\sigma^2 c^2 \|\Gamma^{(t)}\|_F^2 = \eta^2\sigma^2 c^2 \,\text{tr}\left(\Gamma^{(t)\top}\Gamma^{(t)}\right) = \eta^2\sigma^2 c^2 \sum_{j \in [r], k \in [d]} (\Gamma^{(t)})_{ij}^2 \triangleq \eta^2\sigma^2 c^2 v^{(t)}.$$

Since $\text{vec}(\Gamma^{(t)}) \sim N(0, I_{rd})$, $u^{(t)} \sim N(0,1)$ and $v^{(t)} \sim \chi^2_{rd}$. For simplicity, write $d^{(t)} \triangleq \text{dist}(G^{(t)}, \hat{G})$. Now we have the following inequality:

$$d^{(t+1)2} \le D^{(t+1)2} + 2\eta\sigma c D^{(t+1)} u^{(t)} + \eta^2\sigma^2 c^2 v^{(t)}.$$

Let $V \triangleq \max_{t \in [T]} \|g^{(t)} - \partial_G \mathcal{L}_{\text{L}}(G^{(t)})\|_F$. Using Lemma D.1, we obtain

$$d^{(t+1)2} \le (1 - \eta\beta_l)d^{(t)2} + 2\eta\langle H^{(t)}(G^{(t)} - \eta\partial_G \mathcal{L}_{\text{L}}(G^{(t)})) - \hat{G}, \ \partial_G \mathcal{L}_{\text{L}}(G^{(t)}) - g^{(t)}\rangle + \eta^2 V^2$$

$$+ 2\eta\sigma c D^{(t+1)} u^{(t)} + \eta^2\sigma^2 c^2 v^{(t)}, \tag{D.9}$$

which holds for all $t \in [T]$.

We show by induction that the following inequality holds with probability $1 - O(sT^{-1}(n+d)^{-1})$ for any fixed $s \in [T]$:

$$d^{(s)2} \le \min\{\gamma^2, \ 4(1 - \eta\beta_l)^s d^{(0)2} + L\}, \tag{D.10}$$

where $L > 0$ is the solution of $L - C_2\sqrt{L} - C_1^2 = 0$ with

$$C_1 \triangleq \sqrt{\frac{\eta}{\beta_l}\sigma^2 c^2(rd + 14\log^2(T(n+d))) + 7\frac{\eta}{\beta_l}V^2\log(T(n+d))},$$

$$C_2 \triangleq 2\sigma c\sqrt{2\log(T(n+d))\frac{\eta}{\beta_l}} + 2V\sqrt{\log(T(n+d))\frac{\eta}{\beta_l}}.$$

**Step 1.** We start from $s = 1$. From a standard concentration inequality for Gaussian random variables, (see, for example, Proposition 2.5 of Wainwright (2019).)

$$u^{(0)} \leq \sqrt{2 \log(T(n+d))} \tag{D.11}$$

holds with probability at least $1 - T^{-1}(n+d)^{-1}$. From a concentration bound for chi-squared distribution, (see, for example, Lemma 1 of Laurent & Massart (2000).)

$$v^{(0)} \leq rd + 2 \log(T(n+d)) \tag{D.12}$$

holds with probability $1 - cT^{-1}(n+d)^{-1}$ for some universal constant $c > 0$. Note that Lemma D.1 and Cauchy-Schwarz inequality yield

$$
\begin{aligned}
D^{(1)2} &\leq (1 - \eta\beta_l)d^{(0)2} + 2\eta d^{(0)}V + \eta^2 V^2 \\
&\leq (1 - \eta\beta_l)d^{(0)2} + \eta\beta_l d^{(0)2} + 2\frac{\eta}{\beta_l}V^2 \\
&\leq 2(1 - \eta\beta_l)d^{(0)2} + 2\frac{\eta}{\beta_l}V^2,
\end{aligned}
$$

where we used $2xy \leq x^2 + y^2$ in the second inequality and $\eta\beta_l \leq 1/2 \leq 1 - \eta\beta_l$ in the third inequality. Using $\sqrt{x+y} \leq \sqrt{x} + \sqrt{y}$ for $x, y \geq 0$, we further obtain

$$D^{(1)} \leq \sqrt{2}(1 - \eta\beta_l)^{1/2}d^{(0)} + \sqrt{2\frac{\eta}{\beta_l}}V.$$

Combined with equation D.9, D.11 and D.12, we have

$$
\begin{aligned}
d^{(1)2} &\leq D^{(1)2} + 2\eta\sigma c D^{(1)}u^{(0)} + \eta^2\sigma^2 c^2 v^{(0)} \\
&\leq 2(1 - \eta\beta_l)d^{(0)2} + 2\frac{\eta}{\beta_l}V^2 + 2\eta\sigma c\sqrt{2\log(T(n+d))}D^{(1)} + \eta^2\sigma^2 c^2(rd + 2\log(T(n+d))) \\
&\leq 2(1 - \eta\beta_l)d^{(0)2} + 2\frac{\eta}{\beta_l}V^2 + 4\eta\sigma c\sqrt{2\log(T(n+d))}\left((1 - \eta\beta_l)^{1/2}d^{(0)} + \sqrt{\frac{\eta}{\beta_l}}V\right) \\
&\quad + \eta^2\sigma^2 c^2(rd + 2\log(T(n+d))) \\
&\leq 4(1 - \eta\beta_l)d^{(0)2} + 4\frac{\eta}{\beta_l}V^2 + \eta^2\sigma^2 c^2(rd + 10\log(T(n+d))),
\end{aligned}
$$

where we used $2xy \leq x^2 + y^2$ in the last inequality. Notice that

$$
\begin{aligned}
\eta^2\sigma^2 c^2(rd + 10\log(T(n+d))) + 4\frac{\eta}{\beta_l}V^2 &\leq \frac{\eta\sigma^2 c^2}{\beta_l}(rd + 14\log^2(T(n+d))) + 7\frac{\eta}{\beta_l}V^2\log(T(n+d)) \\
&= C_1^2 = L - C_2\sqrt{L} \leq L.
\end{aligned}
$$

From equation D.4, equation D.7 and $L \leq \gamma^2/2$, which will be proved later,

$$4(1 - \eta\beta_l)d^{(0)2} + \eta^2\sigma^2 c^2(rd + 10\log(T(n+d))) \leq 4\frac{\gamma^2}{8} + \frac{\gamma^2}{2} \leq \gamma^2.$$

Therefore, we verify equation D.10 for $s = 1$.

**Step 2.** Fix $s \in [T]$. Suppose that equation D.10 holds for all $t$ satisfying $1 \leq t \leq s - 1$ on the event $E$. Examining the induction steps, we can show that the event $E$ occurs with probability at least $1 - (1 + c)(s - 1)T^{-1}(n+d)^{-1}$. A similar concentration argument for $v^{(t)}$ gives,

$$
\begin{aligned}
d^{(s)2} &\leq (1 - \eta\beta_l)d^{(s-1)2} + 2\eta\langle H^{(s-1)}(G^{(s-1)} - \eta\partial_G\mathcal{L}_{\mathrm{L}}(G^{(s-1)})) - \hat{G}, \partial_G\mathcal{L}_{\mathrm{L}}(G^{(s-1)}) - g^{(s-1)}\rangle + \eta^2 V^2 \\
&\quad + 2\eta\sigma c D^{(s)}u^{(s-1)} + \eta^2\sigma^2 c^2(rd + 2\log(T(n+d)))
\end{aligned} \tag{D.13}
$$

holds with probability at least $1 - (1 + c)(s - 1)T^{-1}(n + d)^{-1} - cT^{-1}(n + d)^{-1}$. Applying equation D.13 repeatedly,

$$
\begin{aligned}
d^{(s)2} \leq{}& (1 - \eta\beta_l)^s d^{(0)2} + \frac{1}{\eta\beta_l}\left\{\eta^2\sigma^2 c^2(rd + 2\log(T(n + d))) + \eta^2 V^2\right\} \\
&+ 2\eta\sigma c \underbrace{\sum_{t'=0}^{s-1}(1 - \eta\beta_l)^{s-t'-1}D^{(t'+1)}u^{(t')}}_{T_1^{(s-1)}} \\
&+ 2\eta\underbrace{\sum_{t'=0}^{s-1}(1 - \eta\beta_l)^{s-t'-1}\langle H^{(t')}(G^{(t')} - \eta\partial_G\mathcal{L}_{\mathrm{L}}(G^{(t')})) - \hat{G},\ \partial_G\mathcal{L}_{\mathrm{L}}(G^{(t')}) - g^{(t')}\rangle}_{T_2^{(s-1)}}.
\end{aligned}
$$

$$(\text{D.14})$$

holds with probability at least $1 - (1 + c)(s - 1)T^{-1}(n + d)^{-1} - cT^{-1}(n + d)^{-1}$.

Before bounding $T_1^{(s-1)}$ and $T_2^{(s-1)}$, we derive a concentration inequality for the following sum:

$$
S_a^{(t-1)} \triangleq \sum_{t'=0}^{t-1} a^{t-t'-1}\langle H^{(t')}(G^{(t')} - \partial_G\mathcal{L}_{\mathrm{L}}(G^{(t')})) - \hat{G},\ \partial_G\mathcal{L}_{\mathrm{L}}(G^{(t')}) - g^{(t')}\rangle,
$$

where $a \in (0, 1)$. Fix $t > 0$. Let $\mathcal{F}^{(t')}$ be a filtration generated from $g^{(0)}, g^{(1)}, \dots, g^{(t')}$. Using Cauchy-Schwarz inequality and Lemma D.1, we obtain

$$
|\langle H^{(t')}(G^{(t')} - \eta\partial_G\mathcal{L}_{\mathrm{L}}(G^{(t')})) - \hat{G},\ \partial_G\mathcal{L}_{\mathrm{L}}(G^{(t')}) - g^{(t')}\rangle| \leq (1 - \eta\beta_l)d^{(t')}V.
$$

From Lemma D.2, $\mathbb{E}[g^{(t)}|\mathcal{F}^{(t-1)}] = \partial_G\mathcal{L}_{\mathrm{L}}(G^{(t)})$. Since $G^{(t')}$ and $\partial_G\mathcal{L}_{\mathrm{L}}(G^{(t')})$ are $\mathcal{F}^{(t'-1)}$-measurable,

$$
\mathbb{E}[\langle H^{(t')}(G^{(t')} - \eta\partial_G\mathcal{L}_{\mathrm{L}}(G^{(t')})) - \hat{G},\ \partial_G\mathcal{L}_{\mathrm{L}}(G^{(t')}) - g^{(t')}\rangle|\mathcal{F}^{(t'-1)}] = 0.
$$

Thus $S_a^{(t-1)}$ is a sum of martingale difference sequence. Using Azuma-Hoeffding bound (See, for example, Corollary 2.20 of Wainwright (2019)), we obtain

$$
|S_a^{(t-1)}| \leq \sqrt{\log(T(n + d))\sum_{t'=0}^{t-1}a^{2t-2t'-2}(1 - \eta\beta_l)d^{(t')2}V^2}.
$$

with probability $1 - O(T^{-1}(n + d)^{-1})$. By a union bound argument,

$$
\max_{t\in[T]}|S_a^{(t-1)}| \leq \sqrt{\log(T(n + d))\sum_{t'=0}^{t-1}a^{2t-2t'-2}(1 - \eta\beta_l)d^{(t')2}V^2} \qquad (\text{D.15})
$$

holds with probability $1 - O((n + d)^{-1})$.

Here we bound the term $T_1^{(s-1)}$, since $u_1^{(0)}, u_2^{(0)}, u_1^{(1)}, u_2^{(1)}, \dots, u_1^{(s-1)}, u_2^{(s-1)}$ are i.i.d. standard normal random variables,

$$
\begin{aligned}
T_1^{(s-1)} &\leq 2\eta\sigma c\sqrt{2\mathrm{Var}(w^{(s-1)})\log(T(n + d))} \\
&\leq 2\eta\sigma c\sqrt{\log(T(n + d))\sum_{t'=0}^{s-1}(1 - \eta\beta_l)^{2s-2t'-2}D^{(t'+1)2}}.
\end{aligned}
$$

We bound $\sum_{t'=0}^{s-1}(1-\eta\beta_l)^{2s-2t'-2}D^{(t'+1)2}$. From Lemma D.1 and equation D.10,

$$\sum_{t'=0}^{s-1}(1-\eta\beta_l)^{2s-2t'-2}D^{(t'+1)2}$$

$$\leq \sum_{t'=0}^{s-1}(1-\eta\beta_l)^{2s-2t'-2}\Big((1-\eta\beta_l)d^{(t')2} + 2\eta\langle H^{(t')}(G^{(t')} - \eta\partial_G\mathcal{L}_L(G^{(t')})) - \hat{G},\ \partial_G\mathcal{L}_L(G^{(t')}) - \eta g^{(t')}\rangle + \eta^2 V^2\Big)$$

$$\leq \sum_{t'=0}^{s-1}4(1-\eta\beta_l)^{2s-t'-1}d^{(0)2} + \sum_{t'=0}^{s-1}(1-\eta\beta_l)^{2s-2t'-1}L + 2\eta S_{(1-\eta\beta_l)^2}^{(s-1)} + \sum_{t'=0}^{s-1}(1-\eta\beta_l)^{2s-2t'-2}\eta^2 V^2.$$

Since $1-\eta\beta_l \leq 1$,

$$\sum_{t'=0}^{s-1}(1-\eta\beta_l)^{4s-4t'-4} \leq \sum_{t'=0}^{s-1}(1-\eta\beta_l)^{3s-3t'-3} \leq \sum_{t'=0}^{s-1}(1-\eta\beta_l)^{2s-2t'-2} \leq \sum_{t'=0}^{s-1}(1-\eta\beta_l)^{s-t'-1} \leq \frac{1}{\eta\beta_l}.$$

$$(D.16)$$

Combined with Lemma D.1, equation D.10, equation D.15 and equation D.16,

$$\sum_{t'=0}^{s-1}(1-\eta\beta_l)^{2s-2t'-2}D^{(t'+1)2}$$

$$\leq 4(1-\eta\beta_l)^s\frac{1}{\eta\beta_l}d^{(0)2} + \frac{L}{\eta\beta_l} + 2\eta\sqrt{\log(T(n+d))\sum_{t'=0}^{s-1}(1-\eta\beta_l)^{4s-4t'-3}d^{(t')2}V^2} + \frac{\eta}{\beta_l}V^2$$

$$\leq 4(1-\eta\beta_l)^s\frac{1}{\eta\beta_l}d^{(0)2} + \frac{L}{\eta\beta_l} + \frac{\eta}{\beta_l}V^2 + 4\eta V\sqrt{\log(T(n+d))\sum_{t'=0}^{s-1}(1-\eta\beta_l)^{4s-3t'-3}d^{(0)2}}$$

$$+ 2\eta V\sqrt{\log(T(n+d))\sum_{t'=0}^{s-1}(1-\eta\beta_l)^{4s-4t'-3}L}$$

$$= 4(1-\eta\beta_l)^s\frac{1}{\eta\beta_l}d^{(0)2} + \frac{L}{\eta\beta_l} + \frac{\eta}{\beta_l}V^2 + 4\eta V\sqrt{\log(T(n+d))(1-\eta\beta_l)^s d^{(0)2}\sum_{t'=0}^{s-1}(1-\eta\beta_l)^{3s-3t'-3}}$$

$$+ 2\eta V\sqrt{\log(T(n+d))\sum_{t'=0}^{s-1}(1-\eta\beta_l)^{4s-4t'-3}L}$$

$$\leq 4(1-\eta\beta_l)^s\frac{1}{\eta\beta_l}d^{(0)2} + \frac{L}{\eta\beta_l} + \frac{\eta}{\beta_l}V^2 + 4\eta V\sqrt{\log(T(n+d))(1-\eta\beta_l)^s d^{(0)2}\frac{1}{\eta\beta_l}}$$

$$+ 2\eta V\sqrt{\log(T(n+d))L\frac{1}{\eta\beta_l}}$$

$$\leq 8(1-\eta\beta_l)^s\frac{1}{\eta\beta_l}d^{(0)2} + 2\frac{L}{\eta\beta_l} + \frac{\eta}{\beta_l}V^2 + 3\eta^2 V^2\log(T(n+d))$$

holds with probability at least $1 - T^{-1}(n+d)^{-1}$, where we used $\sqrt{x+y} \leq \sqrt{x} + \sqrt{y}$ and $2\sqrt{xy} \leq x + y$ for $x, y \geq 0$. Therefore,

$$
\begin{aligned}
T_1^{(s-1)} &\leq 2\eta\sigma c\sqrt{\log(T(n+d))\left(8(1-\eta\beta_l)^s\frac{1}{\eta\beta_l}d^{(0)2} + 2\frac{L}{\eta\beta_l} + \frac{\eta}{\beta_l}V^2 + 3\eta^2V^2\log(T(n+d))\right)} \\
&\leq 4\eta\sigma c\sqrt{2\log(T(n+d))(1-\eta\beta_l)^s\frac{1}{\eta\beta_l}d^{(0)2}} + 2\eta\sigma c\sqrt{2\log(T(n+d))\frac{L}{\eta\beta_l}} \\
&\quad + 2\eta\sigma c\sqrt{\log(T(n+d))\frac{\eta}{\beta_l}V^2} + 2\eta\sigma c\sqrt{3\log^2(T(n+d))\eta^2V^2} \\
&\leq (1-\eta\beta_l)^sd^{(0)2} + 8\sigma^2c^2\frac{\eta}{\beta_l}\log(T(n+d)) + 2\sigma c\sqrt{2\log(T(n+d))\frac{\eta}{\beta_l}L} \\
&\quad + \eta^2\sigma^2c^2\log(T(n+d)) + \frac{\eta}{\beta_l}V^2 + 3\eta^2\sigma^2c^2\log^2(T(n+d)) + \eta^2V^2, \quad\text{(D.17)}
\end{aligned}
$$

where we used $\sqrt{x+y} \leq \sqrt{x} + \sqrt{y}$ for $x, y \geq 0$ and $2xy \leq x^2 + y^2$.

We bound the term $T_2^{(s-1)}$. Using equation D.15 and equation D.10,

$$
\begin{aligned}
T_2^{(s-1)} &= 2\eta S_{1-\eta\beta_l}^{(s-1)} \\
&\leq 2\eta V\sqrt{\log(T(n+d))\sum_{t'=0}^{s-1}(1-\eta\beta_l)^{2s-2t'-1}d^{(t')2}} \\
&\leq 4\eta V\sqrt{\log(T(n+d))\sum_{t'=0}^{s-1}(1-\eta\beta_l)^{2s-t'-1}d^{(0)2}} + 2\eta V\sqrt{\log(T(n+d))\sum_{t'=0}^{s-1}(1-\eta\beta_l)^{2s-2t'-1}L} \\
&\leq 4\eta V\sqrt{\log(T(n+d))(1-\eta\beta_l)^s\frac{1}{\eta\beta_l}d^{(0)2}} + 2\eta V\sqrt{\log(T(n+d))\frac{1}{\eta\beta_l}L} \\
&= 4V\sqrt{\log(T(n+d))(1-\eta\beta_l)^s\frac{\eta}{\beta_l}d^{(0)2}} + 2V\sqrt{\log(T(n+d))\frac{\eta}{\beta_l}L} \\
&\leq 4\frac{\eta}{\beta_l}V^2\log(T(n+d)) + 2(1-\eta\beta_l)^sd^{(0)2} + 2V\sqrt{\log(T(n+d))\frac{\eta}{\beta_l}L}, \quad\text{(D.18)}
\end{aligned}
$$

where we used $\sqrt{x+y} \leq \sqrt{x} + \sqrt{y}$ for $x, y \geq 0$, $2xy \leq x^2 + y^2$. From equation D.14, equation D.17 and equation D.18,

$$
\begin{aligned}
d^{(s)2} &\leq (1-\eta\beta_l)^sd^{(0)2} + \frac{1}{\eta\beta_l}\left\{\eta^2\sigma^2c^2(rd + 2\log(T(n+d))) + \eta^2V^2\right\} \\
&\quad + (1-\eta\beta_l)^sd^{(0)2} + 8\sigma^2c^2\frac{\eta}{\beta_l}\log(T(n+d)) + 2\sigma c\sqrt{2\log(T(n+d))\frac{\eta}{\beta_l}L} \\
&\quad + \eta^2\sigma^2c^2\log(T(n+d)) + \frac{\eta}{\beta_l}V^2 + 3\eta^2\sigma^2c^2\log^2(T(n+d)) + \eta^2V^2 \\
&\quad + 4\frac{\eta}{\beta_l}V^2\log(T(n+d)) + 2(1-\eta\beta_l)^sd^{(0)2} + 2V\sqrt{\log(T(n+d))\frac{\eta}{\beta_l}L} \\
&\leq 4(1-\eta\beta_l)^sd^{(0)2} + \frac{\eta}{\beta_l}\sigma^2c^2(rd + 14\log^2(T(n+d))) + 7\frac{\eta}{\beta_l}V^2\log(T(n+d)) \\
&\quad + \left(2\sigma c\sqrt{2\log(T(n+d))\frac{\eta}{\beta_l}} + 2V\sqrt{\log(T(n+d))\frac{\eta}{\beta_l}}\right)\sqrt{L}
\end{aligned}
$$

holds with probability at least $1 - (1+c)(s-1)T^{-1}(n+d)^{-1} - cT^{-1}(n+d)^{-1} - T^{-1}(n+d)^{-1} = 1 - (1+c)sT^{-1}(n+d)^{-1}$, where we used $2xy \leq x^2 + y^2$ in the third inequality. Note that

$$
d^{(s)2} \leq 4(1-\eta\beta_l)^sd^{(0)2} + L,
$$

since $C_2\sqrt{L} + C_1^2 = L$. Combined with equation D.4 and $L \le \gamma^2/2$, this further gives $d^{(s)2} \le \gamma^2$. Finally, we bound $L$. Solving $L = C_1^2 + C_2\sqrt{L}$ gives

$$
\begin{aligned}
L &= \left(\frac{C_2 + \sqrt{C_2^2 + 4C_1^2}}{2}\right)^2 \le (C_1 + C_2)^2 \le 2C_1^2 + 2C_2^2 \\
&= 2\frac{\eta}{\beta_l}\sigma^2 c^2(rd + 14\log^2(T(n+d))) + 14\frac{\eta}{\beta_l}V^2\log(T(n+d)) \\
&\quad + 2\left(2\sigma c\sqrt{2\log(T(n+d))\frac{\eta}{\beta_l}} + 2V\sqrt{\log(T(n+d))\frac{\eta}{\beta_l}}\right)^2 \\
&\le \frac{\eta\sigma^2 c^2}{\beta_l}\left(2rd + 60\log^2(T(n+d))\right) + 30\frac{\eta}{\beta_l}V^2\log(T(n+d)),
\end{aligned}
$$

where we used $\eta\beta_l \le 1/2$. Note that from equation D.6 and Lemma D.3,

$$
V = \max_{t\in[T]}\|g^{(t)} - \partial_G\mathcal{L}_{\mathrm{L}}(G^{(t)})\|_F \lesssim (\sqrt{r}\|\hat{G}\| + \gamma)L\sqrt{\frac{\log(T(n+d))}{b}} \ll \frac{\gamma}{\sqrt{\log(T(n+d))}}.
$$

Thus

$$
\frac{\eta}{\beta_l}V^2\log(T(n+d)) \le V^2\log(T(n+d)) \le \frac{\gamma^2}{4\cdot 30}.
$$

From equation D.7, we can see that $L \le \gamma^2/2$. Finally, since $4(1-\eta\beta_l)^s d^{(0)2} \le 4\gamma^2/8 = \gamma^2/2$,

$$
d^{(s)2} \le \min\left\{\gamma^2,\ 4(1-\eta\beta_l)^s d^{(0)2} + \frac{\eta\sigma^2 c^2}{\beta_l}\left(2rd + 60\log^2(T(n+d))\right) + 30\frac{\eta}{\beta_l}V^2\right\}
$$

holds with probability $1 - O(sT^{-1}(n+d)^{-1})$ for all $s \in [T]$. This concludes the induction. Again, Lemma D.3 concludes the proof. $\qquad\square$

### D.4 STATISTICAL ERROR BOUND

*Assumption* D.2. Assume that $n \wedge d > r$ and

$$
n \gg \left(\alpha^2 + \frac{1}{\alpha^2}\right)r(r + s_1^{-2}r_e(\Sigma_\xi) + s_2^{-2}r_e(\Sigma_{\tilde{\xi}}))^2\log^3(T(n+d)).
$$

*Assumption* D.3 (Signal-to-noise Ratio). Assume that $s_1^2 \wedge s_2^2 = \Omega(1)$.

*Assumption* D.4 (Signal Condition Number). Assume that $\kappa \triangleq \lambda_{\max}(\Sigma_z)/\lambda_{\min}(\Sigma_z) = O(1)$.

In this section, we let $G^* = [G_1^*, G_2^*]$ be the minimizer of the loss $\mathbb{E}[\mathcal{L}_{\mathrm{L}}(G)]$. Also let $\hat{G} = [\hat{G}_1, \hat{G}_2]$ be the minimizer of the loss $\mathcal{L}_{\mathrm{L}}(G)$. Before going into the proof of Theorem D.1, we introduce lemmas to be used in the proof, which are based on Lemma B.7 in Gao & Ma (2021). Write $\Sigma_{x,\tilde{x}} \triangleq \mathbb{E}[\hat{\Sigma}_{x,\tilde{x}}]$.

**Lemma D.5.** *Suppose that Assumption D.2 holds. Choose $\gamma > 0$ such that*

$$
\gamma \le \min\left\{1,\ \frac{\lambda_r(\hat{\Sigma}_{x,\tilde{x}}) - \lambda_{r+1}(\hat{\Sigma}_{x,\tilde{x}})}{18\alpha(1 + \lambda_1(\hat{\Sigma}_{x,\tilde{x}})/\alpha)^{1/2}}\right\}. \tag{D.19}
$$

*Then, Assumption D.1 holds with*

$$
\beta_u \ge 8\|\hat{\Sigma}_{x,\tilde{x}}\| + 12\alpha,\ \ \beta_l \le \frac{\lambda_r(\hat{\Sigma}_{x,\tilde{x}}) - \lambda_{r+1}(\hat{\Sigma}_{x,\tilde{x}})}{2}.
$$

**Lemma D.6.** *Let $\mathcal{L}'_L(G;\Sigma) := -\operatorname{tr}\big(G_1^\top\Sigma G_2\big) + (\alpha/4)\|GG^\top - I\|_F^2$. Suppose that $\lambda_r(\Sigma) > \lambda_{r+1}(\Sigma)$. Then, the minimizer $\hat{G} = [\hat{G}_1, \hat{G}_2]$ of $\mathcal{L}'_L$ satisfies*

$$
\hat{G}_1 = \frac{1}{\sqrt{2}}V\left(I_r + \frac{1}{\alpha}\Lambda_{[r]}\right)^{1/2}P_{[r]}^\top,\ \ \hat{G}_2 = \frac{1}{\sqrt{2}}V\left(I_r + \frac{1}{\alpha}\Lambda_{[r]}\right)^{1/2}Q_{[r]}^\top,
$$

*where $V \in \mathbb{O}_{r,r}$ is any orthogonal matrix, $\Lambda_{[r]}$ is the top-$r$ singular values of $\Sigma$, and $P_{[r]}$ and $Q_{[r]}$ are the corresponding left and right singular vectors, respectively.*

**Theorem D.2.** *Suppose that Assumptions D.2, D.3 and D.4 hold. Let $G_1^{(T)}$ and $G_2^{(T)}$ be the representation obtained from algorithm 1 under the loss $\mathcal{L}_L(G)$. Suppose that initial representation $G^{(0)}$ satisfy*

$$\text{dist}(G^{(0)}, \hat{G}) \ll \alpha \wedge \frac{1}{\alpha^2}. \tag{D.20}$$

*Choose $c \gg 1 + \alpha$ and $b = \lceil \nu n \rceil$, where $\nu \in (0,1)$ is some constant. If $\eta > 0$ satisfies*

$$\eta \ll \min\left\{ 1 + \frac{1}{\alpha}, \; \frac{1}{\sigma^2(\alpha^3 \vee \alpha^{-1/2})(rd + \log(T(n+d)))} \right\}, \tag{D.21}$$

*then,*

$$\min_{A \in \mathbb{R}^{r \times r}} \|AG_1^{(T)} - G_1^*\|_F \vee \min_{A \in \mathbb{R}^{r \times r}} \|AG_2^{(T)} - G_2^*\|_F$$

$$\lesssim \left(1 - \frac{\eta}{4\kappa}\right)^{T/2} \text{dist}(G^{(0)}, \hat{G}) + \sigma(1+\alpha)\sqrt{\eta(rd + \log(T(n+d)))}$$

$$+ \left(1 + \frac{1}{\alpha}\right)\sqrt{\frac{r(r + s_1^{-2}r_e(\Sigma_\xi) + s_2^{-2}r_e(\Sigma_{\tilde{\xi}}))^2 \log^3(n+d)}{n}}. \tag{D.22}$$

*holds with probability $1 - O((n+d)^{-1})$.*

**Corollary D.1.** *Assume the same conditions as in Theorem D.2. Choose $\eta$ as*

$$\eta = \frac{1}{\sigma\sqrt{T(rd + \log(T(n+d)))}}.$$

*If $T$ satisfies*

$$T \gg \frac{1}{\sigma^2(1 + 1/\alpha)^2(rd + \log(T(n+d)))} \tag{D.23}$$

$$\vee \sigma^2(\alpha^3 \vee \alpha^{-1/2})^2(rd + \log(T(n+d))), \tag{D.24}$$

*then*

$$\min_{A \in \mathbb{R}^{r \times r}} \|AG_1^{(T)} - G_1^*\|_F \vee \min_{A \in \mathbb{R}^{r \times r}} \|AG_2^{(T)} - G_2^*\|_F$$

$$\lesssim \exp\left(-\frac{\sqrt{T}}{8\kappa\sigma\sqrt{rd + \log(T(n+d))}}\right) \text{dist}(G^{(0)}, \hat{G}) + (1+\alpha)\sqrt{\frac{\sigma\sqrt{rd + \log(T(n+d))}}{\sqrt{T}}}$$

$$+ \left(\alpha + \frac{1}{\alpha}\right)\sqrt{\frac{r(r + s_1^{-2}r_e(\Sigma_\xi) + s_2^{-2}r_e(\Sigma_{\tilde{\xi}}))^2 \log^3(n+d)}{n}} \tag{D.25}$$

*holds with probability $1 - O((n+d)^{-1})$.*

*Proof of Corollary D.1.* We directly use Theorem D.2. First, we see that condition D.21 is satisfied from the condition D.24. Note that

$$\sigma(1+\alpha)\sqrt{\eta(rd + \log(T(n+d)))} \lesssim (1+\alpha)\sqrt{\frac{\sigma\sqrt{rd + \log(T(n+d))}}{\sqrt{T}}}. \tag{D.26}$$

The result follows from equation D.26 and equation D.22. $\square$

**Corollary D.2** (Restatement of Theorem 5.1). *Assume the same conditions as in Theorem D.2 and Corollary D.1. Choose $\sigma = C_\sigma \sqrt{T\log(1/\delta)}/(n\epsilon)$ for some universal constant $C_\sigma$. If $T$ satisfies*

$$T \gg \left(\frac{(n\epsilon)}{(1 + 1/\alpha)\sqrt{(rd + \log(T(n+d)))\log(1/\delta)}}\right)^2$$

$$\vee \left(\frac{(\alpha^3 \vee \alpha^{-1/2})\sqrt{(rd + \log(T(n+d)))\log(1/\delta)}}{n\epsilon}\right)^2,$$

*then*

$$\min_{A \in \mathbb{R}^{r \times r}} \|AG_1^{(T)} - G_1^*\|_F \vee \min_{A \in \mathbb{R}^{r \times r}} \|AG_2^{(T)} - G_2^*\|_F$$

$$\lesssim \exp\left(-\frac{n\epsilon}{8\kappa C_\sigma \sqrt{\log(1/\delta)\{rd + \log(T(n+d))\}}}\right) \text{dist}(G^{(0)}, \hat{G}) + (1+\alpha)\frac{(rd + \log(T(n+d)))^{1/4}\log^{1/4}(1/\delta)}{\sqrt{n\epsilon}}$$

$$+ \left(\alpha + \frac{1}{\alpha}\right)\sqrt{\frac{r(r + s_1^{-2}r_e(\Sigma_\xi) + s_2^{-2}r_e(\Sigma_{\tilde{\xi}}))^2 \log^3(n+d)}{n}} \tag{D.27}$$

*holds with probability $1 - O((n+d)^{-1})$.*

Corollary D.2 directly follows from Corollary D.1 with the choice $\sigma \gg \sqrt{T \log(1/\delta)}/(n\epsilon)$.

*Proof of Theorem D.2.* From Lemma D.6, we obtain

$$\hat{G}_1 = \frac{1}{\sqrt{2}}\hat{V}\left(I_r + \frac{1}{\alpha}\hat{\Lambda}_{[r]}\right)^{1/2}\hat{P}_{[r]}^\top, \quad \hat{G}_2 = \frac{1}{\sqrt{2}}\hat{V}\left(I_r + \frac{1}{\alpha}\hat{\Lambda}_{[r]}\right)^{1/2}\hat{Q}_{[r]}^\top,$$

where $\hat{V} \in \mathbb{O}_{r,r}$ is any orthogonal matrix, $\hat{\Lambda}_{[r]} = \text{diag}(\hat{\lambda}_1, \ldots, \hat{\lambda}_r)$ is the top-$r$ singular values of $\hat{\Sigma}_{x,\tilde{x}}$, $\hat{P}_{[r]}$ and $\hat{Q}_{[r]}$ are the left and singular vectors of $\hat{\Sigma}_{x,\tilde{x}}$, respectively. Since $\mathbb{E}[\mathcal{L}_L(G)] = \mathcal{L}'_L(G; \Sigma_{x,\tilde{x}})$, we also obtain

$$G_1^* = \frac{1}{\sqrt{2}}V\left(I_r + \frac{1}{\alpha}\Lambda_{[r]}\right)^{1/2}P_{[r]}^\top, \quad G_2^* = \frac{1}{\sqrt{2}}V\left(I_r + \frac{1}{\alpha}\Lambda_{[r]}\right)^{1/2}Q_{[r]}^\top,$$

where $V \in \mathbb{O}_{r,r}$ is any orthogonal matrix, $\Lambda_{[r]} = \text{diag}(\lambda_1, \ldots, \lambda_r)$ is the top-$r$ singular values of $\Sigma_{x,\tilde{x}}$, $P_{[r]}$ and $Q_{[r]}$ are the left and singular vectors of $\Sigma_{x,\tilde{x}}$, respectively.

We first bound $\min_{A \in \mathbb{R}^{r \times r}} \|A\hat{G}_1 - G_1^*\|_F$. Let $H_P \triangleq \arg\min_{O \in \mathbb{O}_{r,r}} \|O\hat{P}_{[r]}^\top - P_{[r]}^\top\|_F$. Using Theorem 3 from Yu et al. (2015), we have

$$\|H_P\hat{P}_{[r]}^\top - P_{[r]}^\top\|_F \lesssim \frac{(\lambda_1 + 1)}{\lambda_r^2 - \lambda_{r+1}^2}\sqrt{\frac{r(r + s_1^{-2}r_e(\Sigma_\xi) + s_2^{-2}r_e(\Sigma_{\tilde{\xi}}))\log(n+d)}{n}}$$

$$\lesssim \sqrt{\frac{r(r + s_1^{-2}r_e(\Sigma_\xi) + s_2^{-2}r_e(\Sigma_{\tilde{\xi}}))\log(n+d)}{n}}, \tag{D.28}$$

where we used Assumption D.4, $\lambda_{r+1} = 0$ and $\lambda_1 = 1$. Let $A_P := V(I_r + (1/\alpha)\Lambda_{[r]})^{1/2}(I_r + (1/\alpha)\hat{\Lambda}_{[r]})^{-1/2}\hat{V}^{-1}$. Then, from Assumption D.4,

$$\|A_P\|^2 \le \frac{1 + \lambda_1/\alpha}{1 + \hat{\lambda}_r/\alpha} \lesssim 1 \vee \kappa \lesssim 1. \tag{D.29}$$

Moreover,

$$\|A_P\hat{G}_1 - G_1^*\|_F = \left\|V\left(I_r + \frac{1}{\alpha}\Lambda_{[r]}\right)H_P\left(I_r + \frac{1}{\alpha}\hat{\Lambda}_{[r]}\right)^{-1/2}\hat{V}^\top\hat{G}_1 - G_1^*\right\|_F$$

$$= \left\|V\left(I_r + \frac{1}{\alpha}\Lambda_{[r]}\right)(H_P\hat{P}_{[r]}^\top - P_{[r]}^\top)\right\|_F$$

$$\lesssim \left(1 + \frac{1}{\alpha}\right)\sqrt{\frac{r(r + s_1^{-2}r_e(\Sigma_\xi) + s_2^{-2}r_e(\Sigma_{\tilde{\xi}}))\log(n+d)}{n}}, \tag{D.30}$$

where the last inequality follows from equation D.28.

Denote the $j$-th largest singular value of $\Sigma_{x,\tilde{x}}$ and $\hat{\Sigma}_{x,\tilde{x}}$ by $\lambda_j$ and $\hat{\lambda}_j$, respectively. Note that Lemma F.1 and Assumption D.2 gives $\|\hat{\Sigma}_{x,\tilde{x}} - \Sigma_{x,\tilde{x}}\| \ll \|\Sigma_{x,\tilde{x}}\| = 1$ with probability $1 - O((n+d)^{-1})$.

In particular, $\|\hat{\Sigma}_{x,\tilde{x}}\| \leq 2\|\Sigma_{x,\tilde{x}}\| = 2$. Furthermore, from Weyl's inequality, we also have $\max_{j \in [d]} |\hat{\lambda}_j - \lambda_j| \ll (\lambda_1/\lambda_j)\lambda_j \leq \kappa\lambda_j$. Thus, Assumption D.4 gives

$$\hat{\lambda}_r - \hat{\lambda}_{r+1} \geq \lambda_r - \lambda_{r+1} - |\hat{\lambda}_r - \lambda_r| - |\hat{\lambda}_{r+1} - \lambda_{r+1}| \geq \frac{\lambda_r - \lambda_{r+1}}{2} = \frac{1}{2\kappa}.$$

Choose $\gamma > 0$ such that

$$\gamma = \frac{1}{36\kappa(\alpha \vee 1)(1 + 1/(2\alpha))^{1/2}}.$$

Then, $\gamma$ satisfies the condition of Lemma D.5 with probability $1 - O((n + d)^{-1})$. Thus, on this event, Assumption D.1 holds for $\mathcal{L}_L(G)$ with

$$\beta_u \geq 8\hat{\lambda}_1 + 12\alpha, \quad \beta_l \leq \frac{\hat{\lambda}_r - \hat{\lambda}_{r+1}}{2}. \tag{D.31}$$

Choose $\beta_u = 16 + 12\alpha$ and $\beta_l = (\lambda_r - \lambda_{r+1})/4 = 1/(4\kappa)$, which satisfies equation D.31 from the above arguments.

From Lemma F.3,

$$R \lesssim \sqrt{r + s_1^{-2}r_e(\Sigma_\xi)}\sqrt{r + s_2^{-2}r_e(\Sigma_{\tilde{\xi}})}\log(n + d)$$
$$\lesssim (r + s_1^{-2}r_e(\Sigma_\xi) + s_2^{-2}r_e(\Sigma_{\tilde{\xi}}))\log(n + d)$$

holds with probability $1 - O((n+d)^{-1})$. Choose $b = \lceil \nu n \rceil$. Since $\|\hat{G}\|^2 \leq 1 + \|\hat{\Sigma}_{x,\tilde{x}}\|/\alpha \leq 1 + 2/\alpha$,

$$(\sqrt{r}\|\hat{G}\| + 1)R\sqrt{\frac{\log(T(n+d))}{b}} + \gamma\|\hat{\Sigma}_{x,\tilde{x}}\| + \alpha(\|\hat{G}\|^2 + 1)\gamma$$

$$\lesssim \sqrt{r\left(1 + \frac{1}{\alpha}\right)}(r + s_1^{-2}r_e(\Sigma_\xi) + s_2^{-2}r_e(\Sigma_{\tilde{\xi}}))\log(n + d)\sqrt{\frac{\log(T(n+d))}{n}} + 1 + \alpha\left(1 + \frac{1}{\alpha}\right)$$

$$\lesssim \sqrt{\left(1 + \frac{1}{\alpha}\right)}\frac{\sqrt{r}(r + s_1^{-2}r_e(\Sigma_\xi) + s_2^{-2}r_e(\Sigma_{\tilde{\xi}}))\log^{3/2}(n + d)}{\sqrt{n}} + 1 + \alpha$$

$$\lesssim 1 + \alpha, \tag{D.32}$$

where the last inequality follows from Assumption D.2. Also note that

$$\frac{1}{\gamma^2}(\sqrt{r}\|\hat{G}\| + \gamma)^2 R^2 \log(T(n + d))$$

$$\lesssim \kappa^2(\alpha^2 \vee 1)\left(1 + \frac{1}{\alpha}\right)r\left(1 + \frac{1}{\alpha}\right)(r + s_1^{-2}r_e(\Sigma_\xi) + s_2^{-2}r_e(\Sigma_{\tilde{\xi}}))^2 \log^2(n + d)\log(T(n + d))$$

$$\lesssim \left(\alpha^2 + \frac{1}{\alpha^2}\right)r(r + s_1^{-2}r_e(\Sigma_\xi) + s_2^{-2}r_e(\Sigma_{\tilde{\xi}}))^2 \log^3(T(n + d))$$

$$\ll n, \tag{D.33}$$

where the last inequality follows again from Assumption D.2. Choose $c \gg 1 + \alpha$. From equation D.32, equation D.33 and $b = \lceil \nu n \rceil$, we verify that equation D.5 and equation D.6 are satisfied.

From Theorem D.1, if

$$\eta \leq \min\left\{\frac{1}{2(16 + 12\alpha)}, \frac{\lambda_r\gamma^2}{16\sigma^2 c^2(2rd + 60\log(T(n + d)))}\right\},$$

then the following bound holds with probability $1 - O((n + d)^{-1})$:

$$\text{dist}^2(G^{(T)}, \hat{G}) \lesssim (1 - \eta\beta_l)^T \text{dist}^2(G^{(0)}, \hat{G}) + \frac{\eta\sigma^2 c^2}{\beta_l}(rd + \log(T(n + d)))$$

$$+ \frac{\eta}{\beta_l}(\sqrt{r}\|\hat{G}\| + \gamma)^2 R^2 \left(\sqrt{\frac{(1 - b/n)\log(T(n + d))}{b}} + \frac{1}{b}\right)^2.$$

Substituting the values of $c$ and $b$ with a similar argument as in equation D.32 combined with $\sqrt{x+y} \leq \sqrt{x} + \sqrt{y}$ gives

$$\operatorname{dist}(G^{(T)}, \hat{G}) \lesssim (1 - \eta\beta_l)^{T/2} \operatorname{dist}(G^{(0)}, \hat{G}) + \sigma c \sqrt{\frac{\eta}{\beta_l}(rd + \log(T(n+d)))}$$

$$+ \sqrt{\frac{\eta}{\beta_l} r\left(1 + \frac{1}{\alpha}\right)} (r + r_e(\Sigma_\xi) + r_e(\Sigma_{\tilde{\xi}})) \log(n+d) \sqrt{\frac{\log(T(n+d))}{n}}$$

$$\lesssim (1 - \eta\beta_l)^{T/2} \operatorname{dist}(G^{(0)}, \hat{G}) + (1+\alpha)\sigma \sqrt{\frac{\eta}{\beta_l}(rd + \log(T(n+d)))} \tag{D.34}$$

$$+ \sqrt{\frac{\eta}{\beta_l} r\left(1 + \frac{1}{\alpha}\right)} \frac{(r + r_e(\Sigma_\xi) + r_e(\Sigma_{\tilde{\xi}})) \log^{3/2}(T(n+d))}{\sqrt{n}}. \tag{D.35}$$

Finally, note that

$$\min_{A \in \mathbb{R}^{r \times r}} \|AG_1^{(T)} - G_1^*\|_F \leq \min_{A \in \mathbb{R}^{r \times r}} \|AG_1^{(T)} - A_P\hat{G}_1\|_F + \|A_P\hat{G}_1 - G_1^*\|_F$$

$$= \min_{A \in \mathbb{R}^{r \times r}} \|A_P(A_P^{-1}AG_1^{(T)} - \hat{G}_1)\|_F + \|A_P\hat{G}_1 - G_1^*\|_F$$

$$\leq \|A_P\| \min_{O \in \mathbb{O}_{r,r}} \|OG_1^{(T)} - \hat{G}_1\|_F + \|A_P\hat{G}_1 - G_1^*\|_F$$

$$\lesssim \operatorname{dist}(G^{(T)}, \hat{G}) + \|A_P\hat{G}_1 - G_1^*\|_F,$$

where the last inequality follows from equation D.29. Using equation D.30 and equation D.35, we obtain

$$\min_{A \in \mathbb{R}^{r \times r}} \|AG_1^{(T)} - G_1^*\|_F \leq \min_{A \in \mathbb{R}^{r \times r}} \|AG_1^{(T)} - A_P\hat{G}_1\|_F + \|A_P\hat{G}_1 - G_1^*\|_F$$

$$\lesssim (1 - \eta\beta_l)^{T/2} \operatorname{dist}(G^{(0)}, \hat{G}) + \sigma(1+\alpha)\sqrt{\frac{\eta}{\beta_l}(rd + \log(T(n+d)))}$$

$$+ \sqrt{\frac{\eta}{\beta_l} r\left(1 + \frac{1}{\alpha}\right)} \frac{(r + r_e(\Sigma_\xi) + r_e(\Sigma_{\tilde{\xi}})) \log^{3/2}(T(n+d))}{\sqrt{n}}$$

$$+ \left(1 + \frac{1}{\alpha}\right) \sqrt{\frac{r(r + s_1^{-2}r_e(\Sigma_\xi) + s_2^{-2}r_e(\Sigma_{\tilde{\xi}})) \log(n+d)}{n}}$$

$$\lesssim (1 - \eta\beta_l)^{T/2} \operatorname{dist}(G^{(0)}, \hat{G}) + \sigma(1+\alpha)\sqrt{\frac{\eta}{\beta_l}(rd + \log(T(n+d)))}$$

$$+ \left(1 + \frac{1}{\alpha}\right) \sqrt{\frac{r(r + s_1^{-2}r_e(\Sigma_\xi) + s_2^{-2}r_e(\Sigma_{\tilde{\xi}}))^2 \log^3(n+d)}{n}}.$$

A symmetric argument for $\hat{G}_2$ and $G_2^*$ gives the desired result. $\qquad\square$

# E  PROOF OF LEMMAS

*Proof of Lemma D.1.* We first show the following inequality, as in the proof of Lemma 4 in Chi et al. (2019).

$$2\left\langle H\partial_G \mathcal{L}_{\mathrm{L}}(G), HG - \hat{G} \right\rangle \geq \frac{1}{\beta_u} \|\partial_G \mathcal{L}_{\mathrm{L}}(G)\|_F^2 + \beta_l \|HG - \hat{G}\|_F^2. \tag{E.1}$$

Note that $H\partial_G \mathcal{L}_{\mathrm{L}}(G) = \partial_G \mathcal{L}_{\mathrm{L}}(HG)$. Applying Taylor series expansion to $\mathcal{L}_{\mathrm{L}}(\hat{G})$, we obtain

$$\mathcal{L}_{\mathrm{L}}(\hat{G}) = \mathcal{L}_{\mathrm{L}}(HG) - \left\langle H\partial_G \mathcal{L}_{\mathrm{L}}(G), HG - \hat{G} \right\rangle + \frac{1}{2} \operatorname{vec}(HG - \hat{G})^\top \frac{\partial^2 \mathcal{L}_{\mathrm{L}}(\check{G})}{\partial \operatorname{vec}(G) \partial \operatorname{vec}(G)^\top} \operatorname{vec}(HG - \hat{G}),$$

where $\check{G} \triangleq HG + \tau(\hat{G} - HG)$ with some $\tau \in [0, 1]$. We can see that

$$\|\check{G} - \hat{G}\|_F^2 = (1 - \tau)\|HG - \hat{G}\|_F^2 \leq \gamma^2.$$

From equation D.2,

$$\mathcal{L}_{\mathrm{L}}(\hat{G}) \geq \mathcal{L}_{\mathrm{L}}(HG) - \left\langle H\partial_G\mathcal{L}_{\mathrm{L}}(G), HG - \hat{G} \right\rangle + \frac{\beta_l}{2}\|HG - \hat{G}\|_F^2. \tag{E.2}$$

Furthermore, from equation D.1 and equation D.3,

$$\begin{aligned}
\mathcal{L}_{\mathrm{L}}(\hat{G}) - \mathcal{L}_{\mathrm{L}}(HG) &\leq \mathcal{L}_{\mathrm{L}}\left(HG - \frac{1}{\beta_u}\partial_G\mathcal{L}_{\mathrm{L}}(HG)\right) - \mathcal{L}_{\mathrm{L}}(HG) \\
&\leq -\frac{1}{\beta_u}\langle\partial_G\mathcal{L}_{\mathrm{L}}(HG), \partial_G\mathcal{L}_{\mathrm{L}}(HG)\rangle + \frac{\beta_u}{2}\left\|\frac{1}{\beta_u}\partial_G\mathcal{L}_{\mathrm{L}}(HG)\right\|_F^2 \\
&= -\frac{1}{2\beta_u}\|\partial_G\mathcal{L}_{\mathrm{L}}(HG)\|_F^2 = -\frac{1}{2\beta_u}\|\partial_G\mathcal{L}_{\mathrm{L}}(G)\|_F^2, \tag{E.3}
\end{aligned}$$

where the second inequality follows from the characterization of smoothness (Theorem 5.8 of Beck (2017).) From equation E.2 and equation E.3, we obtain

$$\begin{aligned}
-\left\langle H\partial_G\mathcal{L}_{\mathrm{L}}(G), HG - \hat{G}\right\rangle + \frac{\beta_l}{2}\|HG - \hat{G}\|_F^2 &\leq \mathcal{L}_{\mathrm{L}}(\hat{G}) - \mathcal{L}_{\mathrm{L}}(HG) \\
&\leq -\frac{1}{2\beta_u}\|\partial_G\mathcal{L}_{\mathrm{L}}(G)\|_F^2.
\end{aligned}$$

This proves equation E.1.

Next, we bound $\|H\bar{G} - \hat{G}\|_F$. Observe that

$$\begin{aligned}
\|H\bar{G} - \hat{G}\|_F^2 &= \|HG - \hat{G} - \eta H\partial_G\mathcal{L}_{\mathrm{L}}(G)\|_F^2 \\
&= \mathrm{dist}^2(G, \hat{G}) + \eta^2\|\partial_G\mathcal{L}_{\mathrm{L}}(G)\|_F^2 - 2\eta\left\langle\partial_G\mathcal{L}_{\mathrm{L}}(G), HG - \hat{G}\right\rangle \\
&\leq (1 - \eta\beta_l)\,\mathrm{dist}^2(G, \hat{G}) + \eta\left(\eta - \frac{1}{\beta_u}\right)\|\partial_G\mathcal{L}_{\mathrm{L}}(G)\|_F^2 \\
&\leq (1 - \eta\beta_l)\,\mathrm{dist}^2(G, \hat{G}), \tag{E.4}
\end{aligned}$$

where the last inequality follows from $\eta \leq 1/\beta_u$.

Using equation E.4, we further obtain

$$\begin{aligned}
\|H\tilde{G} - \hat{G}\|_F^2 &= \|H\bar{G} - \hat{G} + \eta H(\partial_G\mathcal{L}_{\mathrm{L}}(G) - g)\|_F^2 \\
&= \|H\bar{G} - \hat{G}\|_F^2 + 2\eta\langle H\bar{G} - \hat{G}, \partial_G\mathcal{L}_{\mathrm{L}}(G) - g\rangle + \eta^2\|g - \partial_G\mathcal{L}_{\mathrm{L}}(G)\|_F^2 \\
&\leq (1 - \eta\beta_l)\,\mathrm{dist}^2(G, \hat{G}) + 2\eta\langle H\bar{G} - \hat{G}, \partial_G\mathcal{L}_{\mathrm{L}}(G) - g\rangle + \eta^2\|g - \partial_G\mathcal{L}_{\mathrm{L}}(G)\|_F^2.
\end{aligned}$$

This concludes the proof. $\qquad\square$

*Proof of Lemma D.2.* Observe that

$$\begin{aligned}
\mathbb{E}_{\mathcal{B}}\left[\frac{1}{b}\sum_{i\in\mathcal{B}}x_i\tilde{x}_i^\top\right] &= \frac{1}{b}\frac{1}{\binom{n}{b}}\sum_{\substack{\mathcal{B}'\subset[n] \\ |\mathcal{B}'|=b}}\sum_{i\in\mathcal{B}'}x_i\tilde{x}_i^\top \\
&= \frac{1}{b}\frac{1}{\binom{n}{b}}\sum_{i\in[n]}x_i\tilde{x}_i^\top\sum_{\substack{\mathcal{B}'\subset[n] \\ |\mathcal{B}'|=b}}\mathbf{1}\{i\in\mathcal{B}'\} \\
&= \frac{1}{b}\frac{\binom{n-1}{b-1}}{\binom{n}{b}}\sum_{i\in[n]}x_i\tilde{x}_i^\top \\
&= \frac{1}{n}\sum_{i\in[n]}x_i\tilde{x}_i^\top.
\end{aligned}$$

Similarly,

$$\mathbb{E}_{\mathcal{B}}\left[\frac{1}{b(b-1)}\sum_{\substack{i,j\in\mathcal{B}\\i\neq j}}x_i\tilde{x}_j^\top\right] = \frac{1}{b(b-1)}\frac{1}{\binom{n}{b}}\sum_{\substack{\mathcal{B}'\subset[n]\\|\mathcal{B}'|=b}}\sum_{\substack{i,j\in\mathcal{B}'\\i\neq j}}x_i\tilde{x}_j^\top$$

$$= \frac{1}{b(b-1)}\frac{\binom{n-2}{b-2}}{\binom{n}{b}}\sum_{\substack{i,j\in[n]\\i\neq j}}x_i\tilde{x}_j^\top$$

$$= \frac{1}{n(n-1)}\sum_{\substack{i,j\in[n]\\i\neq j}}x_i\tilde{x}_j^\top.$$

Thus $\mathbb{E}_{\mathcal{B}}[\hat{\Sigma}_{x,\tilde{x},\mathcal{B}}] = \mathbb{E}_{\mathcal{B}}[\hat{\Sigma}_{x,\tilde{x}}]$ and hence

$$\mathbb{E}_{\mathcal{B}}[\mathcal{L}_{\mathrm{L}}(G;\mathcal{B})] = \mathcal{L}_{\mathrm{L}}(G). \tag{E.5}$$

Taking derivative with $G$ in equation E.5 concludes the proof. $\qquad\square$

*Proof of Lemma D.3.* Note that $\partial_G\mathcal{L}_{\mathrm{L}}(G^{(t)};\mathcal{B}^{(t)}) - \partial_G\mathcal{L}_{\mathrm{L}}(G^{(t)};\mathcal{B}^{(t)}) = \partial_G(-\mathrm{tr}(G_1^{(t)}\hat{\Sigma}_{x,\tilde{x}}G_2^{(t)}) + \mathrm{tr}(G_1^{(t)}\hat{\Sigma}_{x,\tilde{x},\mathcal{B}^{(t)}}G_2^{(t)}))$. Thus

$$\begin{aligned}
&\|\partial_G\mathcal{L}_{\mathrm{L}}(G^{(t)};\mathcal{B}^{(t)}) - \partial_G\mathcal{L}_{\mathrm{L}}(G^{(t)};\mathcal{B}^{(t)})\|_F\\
&= \|G_1^{(t)}(\hat{\Sigma}_{x,\tilde{x}} - \hat{\Sigma}_{x,\tilde{x},\mathcal{B}^{(t)}})\|_F + \|G_2^{(t)}(\hat{\Sigma}_{x,\tilde{x}} - \hat{\Sigma}_{x,\tilde{x},\mathcal{B}^{(t)}})^\top\|_F\\
&\leq (\|G_1^{(t)}\|_F + \|G_2^{(t)}\|_F)\|\hat{\Sigma}_{x,\tilde{x}} - \hat{\Sigma}_{x,\tilde{x},\mathcal{B}^{(t)}}\|\\
&\leq (\|\hat{G}\|_F + \mathrm{dist}(G^{(t)},\hat{G}))\|\hat{\Sigma}_{x,\tilde{x}} - \hat{\Sigma}_{x,\tilde{x},\mathcal{B}^{(t)}}\|\\
&\leq (\|\hat{G}\|_F + \gamma)\|\hat{\Sigma}_{x,\tilde{x}} - \hat{\Sigma}_{x,\tilde{x},\mathcal{B}^{(t)}}\|.
\end{aligned}$$

We first bound $\|(1/b)\sum_{i\in\mathcal{B}^{(t)}}x_i\tilde{x}_i^\top - (1/n)\sum_{i\in[n]}x_i\tilde{x}_i^\top\|$. Write $x_i = (x_{i1},\ldots,x_{id_1})$ and $\tilde{x}_i = (\tilde{x}_{i1},\ldots,\tilde{x}_{id_2})$. For any fixed $k\in[d_1]$ and $\ell\in[d_2]$, using Lemma F.2, we have

$$\left|\frac{1}{b}\sum_{i\in\mathcal{B}^{(t)}}x_{ik}\tilde{x}_{il} - \frac{1}{n}\sum_{i\in[n]}x_{ik}\tilde{x}_{il}\right| \leq C\max_{i\in[n]}|x_{ik}\tilde{x}_{il}|\sqrt{\frac{(1-b/n)\log(Trd(n+d))}{b}}$$

$$\leq CR\sqrt{\frac{(1-b/n)\log(T(n+d))}{b}}$$

with probability $1 - O(T^{-1}(rd)^{-1}(n+d)^{-1})$, where $C > 0$ is a universal constant. Note that operator norm of a matrix is bounded by the maximum element of the matrix. By a union bound argument,

$$\left\|\frac{1}{b}\sum_{i\in\mathcal{B}^{(t)}}x_i\tilde{x}_i^\top - \frac{1}{n}\sum_{i\in[n]}x_i\tilde{x}_i^\top\right\| \leq \max_{k\in[d_1],l\in[d_2]}\left|\frac{1}{b}\sum_{i\in\mathcal{B}^{(t)}}x_{ik}\tilde{x}_{il} - \frac{1}{n}\sum_{i\in[n]}x_{ik}\tilde{x}_{il}\right|$$

$$\leq CR\sqrt{\frac{(1-b/n)\log(Trd(n+d))}{b}}$$

holds with probability $1 - O(T^{-1}(n+d)^{-1})$.

Let $\mu_{\mathcal{B}^{(t)}} \triangleq (1/b)\sum_{i\in\mathcal{B}^{(t)}}x_i$ and $\tilde{\mu}_{\mathcal{B}^{(t)}} \triangleq (1/b)\sum_{i\in\mathcal{B}^{(t)}}\tilde{x}_i$. Also let $\mu \triangleq (1/n)\sum_{i\in[n]}x_i$ and $\tilde{\mu} \triangleq (1/n)\sum_{i\in[n]}\tilde{x}_i$. Next we bound $\|\mu_{\mathcal{B}^{(t)}}\tilde{\mu}_{\mathcal{B}^{(t)}}^\top - \mu\tilde{\mu}^\top\|$.

Again from Lemma F.2, for any fixed $j\in[d_1]$,

$$|e_j^\top(\mu_{\mathcal{B}^{(t)}} - \mu)| \leq C'\max_{i\in[n]}|e_j^\top x_i|\sqrt{\frac{(1-b/n)\log(Td_1(n+d))}{b}}$$

holds with probability $1 - O(T^{-1}d_1^{-1}(n+d)^{-1})$, where $C' > 0$ is some universal constant. By a union bound argument, we obtain

$$\|\mu_{\mathcal{B}^{(t)}} - \mu\| \leq C' \max_{i \in [n]} \|x_i\| \sqrt{\frac{(1 - b/n) \log(T(n+d))}{b}}$$

holds with probability $1 - O(T^{-1}(n+d)^{-1})$. Similarly,

$$\|\tilde{\mu}_{\mathcal{B}^{(t)}} - \tilde{\mu}\| \leq C' \max_{i \in [n]} \|\tilde{x}_i\| \sqrt{\frac{(1 - b/n) \log(n+d)}{b}}$$

holds with probability $1 - O((n+d)^{-1})$. Thus,

$$\|\mu_{\mathcal{B}^{(t)}} \tilde{\mu}_{\mathcal{B}^{(t)}}^\top - \mu \tilde{\mu}^\top\| \leq \|\mu - \mu_{\mathcal{B}^{(t)}}\| \|\tilde{\mu}_{\mathcal{B}^{(t)}}\| + \|\mu\| \|\tilde{\mu} - \tilde{\mu}_{\mathcal{B}^{(t)}}\|$$

$$\leq 2C' (\max_{i \in [n]} \|x_i\|)(\max_{i \in [n]} \|\tilde{x}_i\|) \sqrt{\frac{(1 - b/n) \log(T(n+d))}{b}},$$

where we used $\|\tilde{\mu}_{\mathcal{B}^{(t)}}\| \leq \max_{i \in [n]} \|\tilde{x}_i\|$ and $\|\mu\| \leq \max_{i \in [n]} \|x_i\|$. Therefore,

$$\|\hat{\Sigma}_{x,\tilde{x},\mathcal{B}^{(t)}} - \hat{\Sigma}_{x,\tilde{x}}\| = \left\| \frac{1}{b-1} \sum_{i \in \mathcal{B}^{(t)}} x_i \tilde{x}_i^\top - \frac{1}{n-1} \sum_{i \in [n]} x_i \tilde{x}_i^\top - \frac{b}{b-1} \mu_{\mathcal{B}^{(t)}} \tilde{\mu}_{\mathcal{B}^{(t)}}^\top + \frac{n}{n-1} \mu \tilde{\mu}^\top \right\|$$

$$\leq \left\| \frac{1}{b} \sum_{i \in \mathcal{B}^{(t)}} x_i \tilde{x}_i^\top - \frac{1}{n} \sum_{i \in [n]} x_i \tilde{x}_i^\top \right\| + \|\mu_{\mathcal{B}^{(t)}} \tilde{\mu}_{\mathcal{B}^{(t)}}^\top - \mu \tilde{\mu}^\top\|$$

$$+ \left\| \frac{1}{b} \sum_{i \in \mathcal{B}^{(t)}} x_i \tilde{x}_i^\top - \frac{1}{b-1} \sum_{i \in \mathcal{B}^{(t)}} x_i \tilde{x}_i^\top \right\| + \left\| \frac{1}{n} \sum_{i \in [n]} x_i \tilde{x}_i^\top + \frac{1}{n-1} \sum_{i \in [n]} x_i \tilde{x}_i^\top \right\|$$

$$\leq (C + 2C') R \sqrt{\frac{(1 - b/n) \log(T(n+d))}{b}}$$

$$+ \frac{1}{b-1} \max_{i \in \mathcal{B}^{(t)}} \|x_i\| \|\tilde{x}_i\| + \frac{1}{n-1} \max_{i \in [n]} \|x_i\| \|\tilde{x}_i\|$$

$$\leq (C + 2C' + 4) R \left( \sqrt{\frac{(1 - b/n) \log(T(n+d))}{b}} + \frac{1}{b} \right)$$

holds with probability $1 - O(T^{-1}(n+d)^{-1})$. A union bound argument for $t \in [T]$ concludes the proof. $\square$

*Proof of Lemma D.4.* Note that

$$\max_{t \in [T]} \|\partial_G \mathcal{L}_{\mathrm{L}}(G^{(t)}; \mathcal{B})\|_F = \max_{t \in [T]} \|\partial_G \mathcal{L}_{\mathrm{L}}(G^{(t)}; \mathcal{B}) - \partial_G \mathcal{L}_{\mathrm{L}}(\hat{G})\|_F$$

$$\leq \max_{t \in [T]} \underbrace{\|\partial_G \mathcal{L}_{\mathrm{L}}(G^{(t)}; \mathcal{B}) - \partial_G \mathcal{L}_{\mathrm{L}}(G^{(t)})\|_F}_{=:T_1^{(t)}} + \max_{t \in [T]} \underbrace{\|\partial_G \mathcal{L}_{\mathrm{L}}(G^{(t)}) - \partial_G \mathcal{L}_{\mathrm{L}}(\hat{G})\|_F}_{=:T_2^{(t)}}.$$

We can bound the term $T_1^{(t)}$ by Lemma D.3 as

$$\max_{t \in [T]} T_1^{(t)} \lesssim (\sqrt{r} \|\hat{G}\| + \gamma) R \sqrt{\frac{\log(T(n+d))}{b}}, \tag{E.6}$$

which holds with probability $1 - O((n+d)^{-1})$. For the term $T_2$, by triangle inequality and the inequality $\|AB\|_F \leq \|A\| \|B\|_F$ for any matrices $A$ and $B$, we obtain

$$T_2^{(t)} \leq \|(G_2^{(t)} - \hat{G}_2) \hat{\Sigma}_{x,\tilde{x}}^\top\|_F + \|(G_1^{(t)} - \hat{G}_1) \hat{\Sigma}_{x,\tilde{x}}\|_F + \alpha \|G^{(t)} G^{(t)\top} G^{(t)} - \hat{G} \hat{G}^T \hat{G}\|_F + \alpha \|G^{(t)} - \hat{G}\|_F$$

$$\leq \|G^{(t)} - \hat{G}\|_F \|\hat{\Sigma}_{x,\tilde{x}}\| + 3\alpha (\|G^{(t)}\| \vee \|\hat{G}\|)^2 \|G^{(t)} - \hat{G}\|_F$$

$$\lesssim \gamma \|\hat{\Sigma}_{x,\tilde{x}}\| + \alpha (\|\hat{G}\|^2 + \gamma^2) \gamma, \tag{E.7}$$

where we used $\|G^{(t)}\| \leq \|\hat{G}\| + \gamma$. The claim follows from equation E.6 and equation E.7. $\square$

*Proof of Lemma D.5.* The following proof uses the technique from the proof of Lemma B.7 in Gao & Ma (2021). We first derive the bound for the smoothness of $\mathcal{L}_\mathrm{L}$. To this aim, compute the derivatives of $\mathcal{L}_\mathrm{L}$. For notational brevity, we write $\Sigma$ for $\hat{\Sigma}_{x,\tilde{x}}$. Observe that

$$\frac{\partial}{\partial G}\|GG^\top - I_r\|_F^2 = 4GG^\top G - 4G.$$

This implies

$$\frac{\partial}{\partial\,\mathrm{vec}(G)}\|GG^\top - I_r\|_F^2 = \mathrm{vec}(4GG^\top G - 4G) = 4(I_d \otimes GG^\top)\,\mathrm{vec}(G) - 4\,\mathrm{vec}(G). \qquad \text{(E.8)}$$

Write column vectors of $G$ as $G = [a_1, \ldots, a_d] \in \mathbb{R}^{r\times d}$. Define

$$A \triangleq \begin{pmatrix} a_1 a_1^\top & a_2 a_1^\top & \ldots & a_d a_1^\top \\ a_1 a_2^\top & a_2 a_2^\top & \ldots & a_d a_2^\top \\ \vdots & \vdots & \ddots & \vdots \\ a_1 a_d^\top & a_2 a_d^\top & \ldots & a_d a_d^\top \end{pmatrix}.$$

Thus,

$$\frac{\partial\mathcal{L}_\mathrm{L}(G)}{\partial G} = -G\begin{pmatrix} O & \Sigma \\ \Sigma^\top & O \end{pmatrix} + \alpha GG^\top G - \alpha G. \qquad \text{(E.9)}$$

Also, equation E.8 further gives

$$\frac{\partial^2}{\partial\,\mathrm{vec}(G)\partial\,\mathrm{vec}(G)^\top}\|GG^\top - I_r\|_F^2 = 4(I_d \otimes GG^\top) + 4G^\top G \otimes I_r + 4A - 4I_{rd}.$$

Therefore,

$$\frac{\partial^2\mathcal{L}_\mathrm{L}(G)}{\partial\,\mathrm{vec}(G)\partial\,\mathrm{vec}(G)^\top} = -\begin{pmatrix} O & \Sigma \\ \Sigma^\top & O \end{pmatrix} \otimes I_r + \alpha I_d \otimes GG^\top + \alpha G^\top G \otimes I_r + \alpha A - \alpha I_{rd}.$$

From Lemma D.6, $\hat{G} = [\hat{G}_1, \hat{G}_2]$ is given by

$$\hat{G}_1 = \frac{1}{\sqrt{2}}V\left(I_r + \frac{1}{\alpha}\Lambda_{[r]}\right)^{1/2}P_{[r]}^\top, \quad \hat{G}_2 = \frac{1}{\sqrt{2}}V\left(I_r + \frac{1}{\alpha}\Lambda_{[r]}\right)^{1/2}Q_{[r]}^\top,$$

where $V \in \mathbb{O}_{r,r}$ is any orthogonal matrix, $P_{[r]}$ and $Q_{[r]}$ are the top-$r$ right and left singular vectors, respectively, and $\Lambda_{[r]}$ is a diagonal matrix of top-$r$ singular values. We note that $\|\hat{G}\|^2 \le \|\hat{G}_1\|^2 + \|\hat{G}_2\|^2 = (1 + \|\Sigma\|/\alpha)$.

Fix any $Z_1 \in \mathbb{R}^{r\times d_1}$ and $Z_2 \in \mathbb{R}^{r\times d_2}$. Let $Z = [Z_1, Z_2] \in \mathbb{R}^{r\times d}$. Write $z_1 \triangleq \mathrm{vec}(Z_1)$, $z_2 \triangleq \mathrm{vec}(Z_2)$ and $z = \mathrm{vec}(Z)$. Since

$$z^\top\left(\begin{pmatrix} O & \Sigma \\ \Sigma^\top & O \end{pmatrix} \otimes I_r\right)z = 2z_1^\top(\Sigma \otimes I_r)z_2 = 2\,\mathrm{tr}(Z_1^\top Z_2 \Sigma^\top),$$

$$z^\top(I_d \otimes GG^\top)z = \mathrm{tr}(Z^\top GG^\top Z),$$

$$z^\top(G^\top G \otimes I_r)z = \mathrm{tr}(Z^\top ZG^\top G),$$

$$z^\top A z = \sum_{i,j}a_j^\top z_i a_i^\top z_j = \mathrm{tr}(ZG^\top ZG^\top),$$

we obtain

$$z^\top\frac{\partial^2\mathcal{L}_\mathrm{L}(G)}{\partial\,\mathrm{vec}(G)\partial\,\mathrm{vec}(G)^\top}z = -2\,\mathrm{tr}(Z_1^\top Z_2 \Sigma^\top) - \alpha\,\mathrm{tr}(Z^\top Z)$$

$$+ \alpha\,\mathrm{tr}(ZG^\top ZG^\top) + \alpha\,\mathrm{tr}(Z^\top GG^\top Z) + \alpha\,\mathrm{tr}(Z^\top ZG^\top G). \qquad \text{(E.10)}$$

Now suppose that $\|G - \hat{G}\|_F \le \gamma$. By Cauchy-Schwarz inequality,

$$z^\top \frac{\partial^2 \mathcal{L}_\mathrm{L}(G)}{\partial \operatorname{vec}(G) \partial \operatorname{vec}(G)^\top} z \le 2\|Z_1\|_F \|Z_2\|_F \|\Sigma\| + 3\alpha\|Z\|_F^2 \|G\|^2$$

$$\le (2\|\Sigma\| + 3\alpha\|G\|^2)\|Z\|_F^2.$$

From $\|G\|^2 \le (\|\hat{G}\| + \gamma)^2 \le 2\|\hat{G}\|^2 + 2\gamma^2$, $\gamma^2 \le 1$ and $\|\hat{G}\|^2 \le (1 + \|\Sigma\|/\alpha)$,

$$z^\top \frac{\partial^2 \mathcal{L}_\mathrm{L}(G)}{\partial \operatorname{vec}(G) \partial \operatorname{vec}(G)^\top} z \le (8\|\Sigma\| + 12\alpha)\|Z\|_F^2. \tag{E.11}$$

Setting $\beta_u \triangleq 8\|\Sigma\| + 12\alpha$ gives the first result for the smoothness of $\mathcal{L}_\mathrm{L}$.

Next we derive the strong directional convexity of $\mathcal{L}_\mathrm{L}$. Let $\Delta_1 \triangleq HZ_1 - \hat{G}_1$, $\Delta_2 \triangleq HZ_2 - \hat{G}_2$ and $\Delta \triangleq [\Delta_1, \Delta_2] = HZ - \hat{G}$. We bound $\operatorname{vec}(\Delta)^\top \partial^2 \mathcal{L}_\mathrm{L}(G) \operatorname{vec}(\Delta)$ from below. We first deal with the case where $G = \hat{G}$. Since $\hat{G}\hat{G}^\top = (1/\alpha)V\Lambda_{[r]}V^\top + I_r$, equation E.10 gives,

$$\operatorname{vec}(\Delta)^\top \frac{\partial^2 \mathcal{L}_\mathrm{L}(\hat{G})}{\partial \operatorname{vec}(G) \partial \operatorname{vec}(G)^\top} \operatorname{vec}(\Delta)$$

$$= -2\operatorname{tr}\big(\Delta_1^\top \Delta_2 \Sigma^\top\big) - \alpha\operatorname{tr}\big(\Delta^\top \Delta\big)$$

$$+ \alpha\operatorname{tr}\big(\Delta\hat{G}^\top \Delta\hat{G}^\top\big) + \alpha\operatorname{tr}\big(\Delta^\top \hat{G}\hat{G}^\top \Delta\big) + \alpha\operatorname{tr}\big(\Delta^\top \Delta\hat{G}^\top \hat{G}\big)$$

$$= \underbrace{\operatorname{tr}\big(\Delta^\top V\Lambda_{[r]}V^\top \Delta\big) + \alpha\operatorname{tr}\big(\Delta^\top \Delta\hat{G}^\top \hat{G}\big) - 2\operatorname{tr}\big(\Delta_1^\top \Delta_2 \Sigma^\top\big)}_{=:T_1} + \underbrace{\alpha\operatorname{tr}\big(\Delta\hat{G}^\top \Delta\hat{G}^\top\big)}_{=:T_2}.$$

We first bound the term $T_1$. Note

$$\hat{G}^\top \hat{G} = \frac{1}{2}\begin{pmatrix} P_{[r]} \\ Q_{[r]} \end{pmatrix}\begin{pmatrix} P_{[r]} \\ Q_{[r]} \end{pmatrix}^\top + \frac{1}{2\alpha}\begin{pmatrix} P_{[r]}\Lambda_{[r]}P_{[r]}^\top & P_{[r]}\Lambda_{[r]}Q_{[r]}^\top \\ Q_{[r]}\Lambda_{[r]}P_{[r]}^\top & Q_{[r]}\Lambda_{[r]}Q_{[r]}^\top \end{pmatrix}. \tag{E.12}$$

Notice that the first term is positive semi-definite. Hence,

$$\alpha\operatorname{tr}\big(\Delta^\top \Delta\hat{G}^\top \hat{G}\big) \ge \frac{1}{2}\operatorname{tr}\big(\Delta_1^\top \Delta_1 P_{[r]}\Lambda_{[r]}P_{[r]}^\top\big) + \operatorname{tr}\big(\Delta_1^\top \Delta_2 Q_{[r]}\Lambda_{[r]}P_{[r]}^\top\big) + \frac{1}{2}\operatorname{tr}\big(\Delta_2^\top \Delta_2 Q_{[r]}\Lambda_{[r]}Q_{[r]}^\top\big)$$

$$\ge 2\operatorname{tr}\big(\Delta_1^\top \Delta_2 Q_{[r]}\Lambda_{[r]}P_{[r]}^\top\big),$$

where we used $\operatorname{tr}(AB) \le \|A\|_F \|B\|_F \le (1/2)\operatorname{tr}\big(A^\top A\big) + (1/2)\operatorname{tr}\big(B^\top B\big)$, which follows from Cauchy-Schwarz inequality and $2xy \le x^2 + y^2$. Let the SVD of $\Sigma$ be $\Sigma = P_{[r]}\Lambda_{[r]}Q_{[r]}^\top + P_\perp \Lambda_\perp Q_\perp^\top$, where $P_\perp$, $Q_\perp$ are the right and left singular vectors except top-$r$ singlar vectors, respectively, and $\Lambda_\perp$ is a diagonal matrix of remaining singular values. Observe that

$$\alpha\operatorname{tr}\big(\Delta^\top \Delta\hat{G}^\top \hat{G}\big) - 2\operatorname{tr}\big(\Delta_1^\top \Delta_2 \Sigma^\top\big)$$

$$\ge 2\operatorname{tr}\big(\Delta_1^\top \Delta_2 Q_{[r]}\Lambda_{[r]}P_{[r]}^\top\big) - 2\operatorname{tr}\big(\Delta_1^\top \Delta_2 Q_{[r]}\Lambda_{[r]}P_{[r]}^\top\big) - 2\operatorname{tr}\big(\Delta_1^\top \Delta_2 Q_\perp \Lambda_\perp P_\perp^\top\big)$$

$$= -2\operatorname{tr}\big(\Delta_1^\top \Delta_2 Q_\perp \Lambda_\perp P_\perp^\top\big).$$

Thus,

$$T_1 \ge \operatorname{tr}\big(\Delta^\top V\Lambda_{[r]}V^\top \Delta\big) - 2\operatorname{tr}\big(\Delta_1^\top \Delta_2 Q_\perp \Lambda_\perp P_\perp^\top\big)$$

$$\ge \lambda_r\|\Delta\|_F^2 - \operatorname{tr}\big(\Delta_2 Q_\perp \Lambda_\perp Q_\perp^\top \Delta_2^\top\big) - \operatorname{tr}\big(\Delta_1 P_\perp \Lambda_\perp P_\perp^\top \Delta_1^\top\big)$$

$$\ge (\lambda_r - \lambda_{r+1})\|\Delta\|_F^2,$$

where we used $\operatorname{tr}(AB) \le (1/2)\operatorname{tr}\big(A^\top A\big) + (1/2)\operatorname{tr}\big(B^\top B\big)$ again.

Next, we show $T_2 \ge 0$. Suppose that $HZ\hat{G}^\top$ is symmetric. Then,

$$\operatorname{tr}\big(\Delta\hat{G}^\top \Delta\hat{G}^\top\big) = \operatorname{tr}\big((HZ - \hat{G})\hat{G}^\top (HZ - \hat{G})\hat{G}^\top\big)$$

$$= \operatorname{tr}\big(HZ\hat{G}^\top HZ\hat{G}^\top\big) - 2\operatorname{tr}\big(\hat{G}\hat{G}^\top HZ\hat{G}^\top\big) + \operatorname{tr}\big(\hat{G}\hat{G}^\top \hat{G}\hat{G}^\top\big)$$

$$= \operatorname{tr}\big((HZ\hat{G}^\top - \hat{G}\hat{G}^\top)^\top (HZ\hat{G}^\top - \hat{G}\hat{G}^\top)\big) \ge 0.$$

Thus, we only need to show that $HZ\hat{G}^\top$ is symmetric. Recall that $H \triangleq \arg\min_{O \in \mathbb{O}_{r,r}} \|OZ - \hat{G}\|_F^2$. This gives

$$\|HZ - \hat{G}\|_F^2 \leq \|H'Z - \hat{G}\|_F^2 \tag{E.13}$$

for all $H' \in \mathbb{O}_{r,r}$. Let the SVD of $Z\hat{G}^\top$ be $Z\hat{G}^\top = UCV^\top$, where $U, V \in \mathbb{O}_{r,r'}$ are positive definite matrices and $C \in \mathbb{R}^{r'}$ is a diagonal matrix. Write the orthogonal matrices of $U$ and $V$ as $U_\perp \in \mathbb{O}_{r,r-r'}$ and $V_\perp \in \mathbb{O}_{r,r-r'}$, respectively. From equation E.13,

$$\mathrm{tr}\left(V^\top H'UC\right) = \sum_{j \in [r']} (V^\top H'U)_{j,j}(C)_{j,j} \leq \mathrm{tr}\left(V^\top HUC\right) \quad \forall H' \in \mathbb{O}_{r,r}. \tag{E.14}$$

The inequality in equation E.14 holds if and only if $V^\top HU = I_r$. Now we decompose $HU$ as $HU = VV^\top HU + V_\perp V_\perp^\top HU = V + V_\perp V_\perp^\top HU$. The fact that $V, HU \in \mathbb{O}_{r,r'}$ yields $HU = V$. Thus $HZ\hat{G}^\top = HUCV^\top = VCV^\top$ and hence $HZ\hat{G}^\top$ is symmetric.

In summary, we showed that

$$\mathrm{vec}(\Delta)^\top \frac{\partial^2 \mathcal{L}_\mathrm{L}(\hat{G})}{\partial \mathrm{vec}(G)\partial \mathrm{vec}(G)^\top} \mathrm{vec}(\Delta) \geq (\lambda_r(\Sigma) - \lambda_{r+1}(\Sigma))\|\Delta\|_F^2. \tag{E.15}$$

Next, we prove that $\mathrm{vec}(\Delta)^\top \partial^2\mathcal{L}_\mathrm{L}(G)\mathrm{vec}(\Delta)$ is close to $\mathrm{vec}(\Delta)^\top \partial^2\mathcal{L}_\mathrm{L}(\hat{G})\mathrm{vec}(\Delta)$ under assumption $\|G - \hat{G}\|_F \leq \gamma$. Observe that

$$\mathrm{vec}(\Delta)^\top \frac{\partial^2 \mathcal{L}_\mathrm{L}(G)}{\partial \mathrm{vec}(G)\partial \mathrm{vec}(G)^\top} \mathrm{vec}(\Delta) - \mathrm{vec}(\Delta)^\top \frac{\partial^2 \mathcal{L}_\mathrm{L}(\hat{G})}{\partial \mathrm{vec}(G)\partial \mathrm{vec}(G)^\top} \mathrm{vec}(\Delta)$$
$$= \alpha \,\mathrm{tr}\left(\Delta G^\top \Delta G^\top\right) + \alpha \,\mathrm{tr}\left(\Delta^\top GG^\top \Delta\right) + \alpha \,\mathrm{tr}\left(\Delta^\top \Delta G^\top G\right)$$
$$- \alpha \,\mathrm{tr}\left(\Delta \hat{G}^\top \Delta \hat{G}^\top\right) - \alpha \,\mathrm{tr}\left(\Delta^\top \hat{G}\hat{G}^\top \Delta\right) - \alpha \,\mathrm{tr}\left(\Delta^\top \Delta \hat{G}^\top \hat{G}\right).$$

Using triangle inequality multiple times, we obtain

$$\left| \mathrm{vec}(\Delta)^\top \frac{\partial^2 \mathcal{L}_\mathrm{L}(G)}{\partial \mathrm{vec}(G)\partial \mathrm{vec}(G)^\top} \mathrm{vec}(\Delta) - \mathrm{vec}(\Delta)^\top \frac{\partial^2 \mathcal{L}_\mathrm{L}(\hat{G})}{\partial \mathrm{vec}(G)\partial \mathrm{vec}(G)^\top} \mathrm{vec}(\Delta) \right|$$

$$\leq 3\alpha\|\Delta\|_F^2\|G - \hat{G}\|_F(\|G\| + \|\hat{G}\|)$$
$$\leq 3\alpha\|\Delta\|_F^2\|G - \hat{G}\|_F(\|G - \hat{G}\| + 2\|\hat{G}\|)$$
$$\leq 3\alpha\|\Delta\|_F^2\gamma\left(1 + 2\left(1 + \frac{\|\Sigma\|}{\alpha}\right)^{1/2}\right)$$
$$\leq 9\alpha\|\Delta\|_F^2\gamma\left(1 + \frac{\|\Sigma\|}{\alpha}\right)^{1/2}$$
$$\leq \frac{\lambda_r(\Sigma) - \lambda_{r+1}(\Sigma)}{2}\|\Delta\|_F^2, \tag{E.16}$$

where the second last inequality follows from the assumption and $\|G - \hat{G}\| \leq \|G - \hat{G}\|_F \leq \gamma \leq 1$, and the last inequality follows from equation D.19. Combining equation E.15 and equation E.16 gives

$$\mathrm{vec}(\Delta)^\top \frac{\partial^2 \mathcal{L}_\mathrm{L}(G)}{\partial \mathrm{vec}(G)\partial \mathrm{vec}(G)^\top} \mathrm{vec}(\Delta) \geq \frac{\lambda_r(\Sigma) - \lambda_{r+1}(\Sigma)}{2}\|\Delta\|_F^2.$$

Setting $\beta_l \triangleq (\lambda_r(\Sigma) - \lambda_{r+1}(\Sigma))/2$ concludes the proof. □

*Proof of Lemma D.6.* Here we derive the minimizer of $\mathcal{L}'_\mathrm{L}$. Let the singular value decomposition of $\Sigma$ be $\Sigma = P\Lambda Q^\top$, where $\Lambda \in \mathbb{R}^{d_1 \times d_2}$ is a diagonal matrix and $P = (p_1, \ldots, p_{d_1}) \in \mathbb{O}_{d_1,d_1}$ and $Q = (q_1, \ldots, q_{d_2}) \in \mathbb{O}_{d_2,d_2}$ are orthogonal matrices. By setting equation E.9 to be 0, we obtain

$$G\begin{pmatrix} O & \Sigma \\ \Sigma^\top & O \end{pmatrix} = \alpha(GG^\top - I_r)G.$$

Equivalently,

$$G_1\Sigma = \alpha(GG^\top - I_r)G_2, \tag{E.17}$$

$$G_2\Sigma^\top = \alpha(GG^\top - I_r)G_1. \tag{E.18}$$

Multiplying $\Sigma$ from the right in equation E.18, and substituting with equation E.17 gives

$$G_2\Sigma^\top\Sigma = \alpha(GG^\top - I_r)G_1\Sigma = \alpha(GG^\top - I_r)^2 G_2.$$

Thus the right singular vectors of $G_2$ is aligned with some $r$ column vectors of $Q$. Given an indices set $J = \{j_1,\ldots,j_r\}$, we write $\Lambda_J \triangleq \mathrm{diag}((\Lambda)_{j_1,j_1},\ldots,(\Lambda)_{j_r,j_r})$, $P_J \triangleq (p_{j_1},\ldots,p_{j_r})$ and $Q_J \triangleq (q_{j_1},\ldots,q_{j_r})$. We decompose $G_2$ by SVD as $G_2 = V_2 C_2 U_2^\top$, where $U_2 = Q_I$ for some $I = (i_1,\ldots,i_r) \subset [d_1]$ and $C_2 \in \mathbb{R}^{r\times r}$ is a diagonal matrix, and $V_2 \in \mathbb{O}_{r,r}$. Similarly, we decompose $G_1$ as $G_1 = V_1 C_1 U_1^\top$, where $U_1 = P_{I'}$ for some $I' = (i'_1,\ldots,i'_r) \subset [d_2]$, $C_1 \in \mathbb{R}^{r\times r}$ is a diagonal matrix, and $V_1 \in \mathbb{O}_{r,r}$.

From equation E.17, we obtain

$$V_1 C_1 \Lambda_{I'} Q_{I'}^\top = \alpha(GG^\top - I_r)V_2 C_2 Q_I^\top.$$

Thus $Q_{I'} = Q_I H_Q$ for some $H_Q \in \mathbb{O}_{r,r}$. Without loss of generality, we set $I = I'$.

Since the term $-\mathrm{tr}\left(G_1\hat{\Sigma}G_2^\top\right) = -\mathrm{tr}\left(V_1 C_1 \Lambda_I C_2 V_2^\top\right)$ in the loss function is minimized when $V_1 = V_2$, whereas the penalty term $\Pi(G_1, G_2)$ is invariant under the change of $V_1$ and $V_2$, we obtain $V_1 = V_2$. In summary, from equation E.17 and equation E.18, we have

$$V_1 C_1 \Lambda_I Q_I^\top = \alpha V_1(C_1^2 + C_2^2 - I_r)C_2 Q_I^\top,$$

$$V_1 C_2 \Lambda_I P_I^\top = \alpha V_1(C_1^2 + C_2^2 - I_r)C_1 P_I^\top.$$

Thus

$$C_1\Lambda_I = \alpha(C_1^2 + C_2^2 - I_r)C_2, \quad C_2\Lambda_I = \alpha(C_1^2 + C_2^2 - I_r)C_1. \tag{E.19}$$

Fix any $j \in [r]$. Suppose that $j$-th entry of $C_1$ is 0. Then, from equation E.19, $j$-th entry of $C_2$ must be 0. Now the loss function can be written as

$$\mathcal{L}_{\mathrm{L}} = -\mathrm{tr}(C_1\Lambda_I C_2) + \frac{\alpha}{4}\|C_1^2 + C_2^2 - I_r\|_F^2.$$

Note that we can make the loss smaller by slightly increasing $j$-th diagonal entry of $C_1$ and $C_2$. This implies that $j$-th diagonal entry of $C_1$ and $C_2$ cannot be 0 when $G_1$ and $G_2$ are the solution to the minimization of the loss $\mathcal{L}_{\mathrm{L}}(G)$. Since $j$ is arbitrary, we can show that $C_1 = C_2 = (1/\sqrt{2})(I_r + (1/\alpha)\Lambda_I)^{1/2}$.

Next we show $I = [r]$. To see this, note that

$$\mathcal{L}_{\mathrm{L}} = -\frac{1}{2}\mathrm{tr}\left(\left(I_r + \frac{1}{\alpha}\Lambda_I\right)^{1/2}\Lambda_I\left(I_r + \frac{1}{\alpha}\Lambda_I\right)^{1/2}\right) + \frac{\alpha}{4}\mathrm{tr}((C_1^2 + C_2^2 - I_r)^2)$$

$$= -\frac{1}{2}\sum_{i\in I}\left(\lambda_i + \frac{\lambda_i^2}{\alpha}\right) + \frac{1}{4\alpha}\sum_{i\in I}\lambda_i^2$$

$$= -\frac{1}{2}\sum_{i\in I}\left(\lambda_i + \frac{\lambda_i^2}{2\alpha}\right),$$

which is minimized if and only if $I = [r]$ due to the assumption $\lambda_r(\Sigma) > \lambda_{r+1}(\Sigma)$. Finally, $\hat{G}$ is given by

$$\hat{G}_1 = \frac{1}{\sqrt{2}}V\left(I_r + \frac{1}{\alpha}\Lambda_{[r]}\right)^{1/2}P_{[r]}^\top, \quad \hat{G}_2 = \frac{1}{\sqrt{2}}V\left(I_r + \frac{1}{\alpha}\Lambda_{[r]}\right)^{1/2}Q_{[r]}^\top, \tag{E.20}$$

where $V \in \mathbb{O}_{r,r}$ is any orthogonal matrix. $\qquad\square$

# F  AUXILIARY RESULTS

Here we define Orlicz norm of a random variable $X$ as $\|X\|_{\psi_2} \triangleq \inf\{c > 0 : \mathbb{E}[e^{X^2/c^2}] \leq 2\}$.

*Assumption* F.1. Let $X$ and $\tilde{X}$ be mean zero random vectors taking values in $\mathbb{R}^{d_1}$ and $\mathbb{R}^{d_2}$, respectively. Assume that there exists some constants $C_1, C_2 > 0$ satisfying that $\mathbb{E}[(u^\top X)^2] \geq C_1 \|u^\top X\|_{\psi_2}^2$ holds for any $u \in \mathbb{R}^{d_1}$, and that $\mathbb{E}[(v^\top \tilde{X})^2] \geq C_2 \|v^\top \tilde{X}\|_{\psi_2}^2$ holds for any $v \in \mathbb{R}^{d_2}$.

We borrow the proposition from Nakada et al. (2023) for bounding the distance between sample cross-covariance matrix and population cross-covariance matrix.

**Lemma F.1** (Proposition 9.1 from Nakada et al. (2023)). *Suppose that Assumption F.1 holds. Let* $(X_1, \tilde{X}_1), \ldots, (X_n, \tilde{X}_n)$ *be independent copies of* $(X, \tilde{X})$. *Let* $\hat{\Sigma}_{X,\tilde{X}} \triangleq (1/n) \sum_{i=1}^n X_i \tilde{X}_i^\top$. *Then, there exists some constant* $C = C(C_1, C_2) > 0$ *such that with probability at least* $1 - e^{-t}$,

$$\|\hat{\Sigma}_{X,\tilde{X}} - \mathbb{E}[\hat{\Sigma}_{X,\tilde{X}}]\|$$

$$\leq C\left[(\operatorname{tr}(\Sigma_{\tilde{X}})\|\Sigma_X\| \vee \operatorname{tr}(\Sigma_X)\|\Sigma_{\tilde{X}}\|)^{1/2}\sqrt{\frac{t + \log(d_1 + d_2)}{n}} \vee (\operatorname{tr}(\Sigma_X)\operatorname{tr}(\Sigma_{\tilde{X}}))^{1/2}\frac{t + \log(d_1 + d_2)}{n}\right]$$

*holds for all* $t > 0$.

**Lemma F.2** (Corollary 2.5 from Bardenet & Maillard (2015)). *Let* $X_1, \ldots, X_n \in \mathbb{R}$ *be fixed numbers. Let* $\mathcal{B} \subset [n]$ *be a random batch of size* $b$ *from* $[n]$ *without replacement. Then,*

$$\left|\frac{1}{b}\sum_{i \in \mathcal{B}} X_i - \frac{1}{n}\sum_{i \in [n]} X_i\right| \lesssim \max_{i \in [n]} |X_i| \sqrt{\frac{(1 - b/n)\log(1/\gamma)}{b}}$$

*holds with probability* $1 - \gamma$.

**Lemma F.3** (Modification of Lemma C.1 from Nakada et al. (2023)). *Suppose Assumptions D.4 and D.3 hold. Then, the following inequalities hold with probability* $1 - O((n + d)^{-1})$:

$$\max_{i \in [n]} \|x_i\| \leq C_1 \|\Sigma_z\|^{1/2}\sqrt{(r + s_1^{-2} r_e(\Sigma_\xi))\log(n + d)},$$

$$\max_{i \in [n]} \|\tilde{x}_i\| \leq C_2 \|\Sigma_{\tilde{z}}\|^{1/2}\sqrt{(r + s_2^{-2} r_e(\Sigma_{\tilde{\xi}}))\log(n + d)},$$

*where* $C_1' = C_1'(\sigma, s_1), C_2' = C_2'(\sigma, s_2)$ *are some constants.*

