# OpenReview forum: "Safeguarding Data in Multimodal AI: A Differentially Private Approach to CLIP Training"
_ICLR.cc/2024/Conference — Submitted to ICLR 2024_

### Official Review · Reviewer_68X7 · 2023-10-31

**Soundness:** 3 good
**Presentation:** 1 poor
**Contribution:** 1 poor
**Rating:** 1
**Confidence:** 4

**Summary:**

The authors propose a CLIP fine-tuning technique that is DP. The idea is to maintain the CLIP contrastive training also at fine-tuning time: to circumvent the issue of having to produce per-sample gradients, the authors propose to apply the gaussian mechanism on the per-batch gradients. The authors make sure this is DP, and carry out an empirical and theoretical analysis about it.

####
POST-REBUTTAL: In lack of any response addressing my major concerns, I decrease my score.

**Strengths:**

- The literature of DP in multi-modality is lacking, and therefore the work could be interesting

**Weaknesses:**

- There are several factually inappropriate usages of the notion of DP. DP is not "incorporated" in a model or multimodality (as the authors mention in different ways a few times throughout the paper), DP is a property of a randomised algorithm (in this context, the training algorithm that produces the distribution of models, not the model).
- Proposition 3.1 is a trivial consequence of the basic theorems of DP, and Algorithm 1 is a simple modification of DP-SGD (that is already available in standard DP-training libraries). The only interesting aspect is adapting the training procedure to contrastive learning, which is trivial. Therefore the novelty of these components of the paper is negligible.
- It is unclear why the zero-shot prediction of CLIP is not used as a baseline (as it would be equivalent to epsilon = 0). Furthermore, the authors are neglecting parameter efficient fine-tuning baselines, for instance like [1]. The description of the baselines is lacking and quite confusing: it's not clear why the authors have two DP-SGD baselines, and what they're exactly updating during training.
- The experimental analysis focuses exclusively on known computer vision datasets that do not differ from the training distribution of CLIP. It would be useful if following the suggestions of [2,3] the authors could present results in settings with low data regimes and with significant distribution shift with respect to the pre-training set, which represents a more realistic application setting.  Furthermore, the authors deliberately avoid settings where DP is known to be hard due to the relatively low amount of training data per class (e.g. CIFAR-100/ImageNet).
- Not accounting for the privacy loss incurred in tuning the hyperparameters is a problem. Simply because baselines have done that, it doesn't justify the authors from reporting inflated numbers. I would recommend the authors to at least assume the availability of some public data that is kept out of training and evaluations and to run all the baselines fairly in this setting.
- BLIP is definitely no more state-of-the-art as the authors claim. Several more sophisticated VLMs have been released since BLIP, e.g. LLaVa 1.5.
- There are several grammatical errors and typos, and the overall presentation is a bit poor. Sections 5 onwards feel disconnected from the rest of the work.

[1] https://arxiv.org/pdf/2110.05679.pdf
[2] https://arxiv.org/pdf/2212.06470.pdf
[3] https://arxiv.org/abs/2306.03962

**Questions:**

- CLIP is relatively old. Novel variants of CLIP may yield zero-shot (epsilon = 0) performance that is higher and might reduce the room for improvements achievable under DP constraints. Could the authors consider the latest and strongest CLIP-inspired architectures?
- Since the gaussian mechanism is not applied at a sample level but at a mini-batch level, are the authors making a fair comparison between the $\epsilon$ values of their technique and the baselines?
- Could the authors show the result of applying the contrastive clip fine-tuning for $\epsilon=\infty$? To know what's the upper bound on accuracy of the DP variant.
- Why the baselines selection varies based on the choice of the dataset?

---

> ### Author Response · Authors · 2023-11-23
>
> Q1: There are several factually inappropriate usages of the notion of DP
>
> A1: Thank you for the suggestion. We would like to note that several papers (from distinguished ML researchers) in the literature also use the wording “incorporate DP” [4,5]. In light of your comment, we will rephrase it to “incorporate privacy” for better description of our method.
>
> Q2: Proposition 3.1 is a trivial consequence of the basic theorems of DP
>
> A2: We would like to clarify that our Proposition 3.1 states that the Gaussian mechanism applied on minibatch level with this calibrated noise satisfies $(\epsilon,\delta)$-DP for per-sample. We acknowledge that the proof of this proposition is straightforward, so we named it as Proposition 3.1 rather than Theorem 3.1 in the original paper. We also would like to emphasize that our contribution is to demonstrate the efficiency of this simple approach for CLIP training that can achieve high accuracy while maintaining DP, supported by both our theory and experiments. The theoretical analysis in Section 5 is quite challenging, as we dealt with ***non-decomposable*** CLIP loss function. The technical tools employed in the existing literature to study DP-SGD do not directly apply to our proposed DP-CLIP. The non-decomposable loss required us to develop novel technical tools for theoretical analysis. In particular, we developed a novel concentration argument to provide the convergence analysis. This new analysis provides theoretical guarantees that per-batch clipping efficiently achieves high accuracy while maintaining DP. We consider this to be one of the major contributions of our work.
>
> Q3: It is unclear why the zero-shot prediction of CLIP is not used as a baseline. Furthermore, the authors are neglecting parameter efficient fine-tuning baselines, for instance like [1]
>
> A3: Thank you for your suggestion. We added both zero-shot prediction results and non-private training results in the universal response. We note that our method is proposed for DP-CLIP training, while [1] focuses on differentially private fine-tuning that optimizes some task-specific loss. Thus their method is not a natural baseline to compare with. We emphasize that our proposed method uniquely addresses the DP training of contrastive vision-language models, not only limited to DP image classifications. From this point, there are no “fair” baselines to compare with. We specifically chose these benchmarks because they demonstrate State-of-the-Art results in image classification tasks on these datasets. Our intention is not to compare with their methods, but rather to evaluate our proposed approach against the established performance standards. As image classification is not one of the tasks in [1], it is not included in our baselines.

---

> ### Author Response · Authors · 2023-11-23
>
> Q4: [...] known computer vision datasets that do not differ from the training distribution of CLIP. [...] results in settings with low data regimes and with significant distribution shift with respect to the pre-training set, following the suggestions of [2,3]
>
> A4: Thank you for your comments! Following your suggestions, we ran our experiments on CIFAR-100. The mean and standard deviation of 10 trials are reported in the universal response. Following your suggestions, we also ran our experiments on a dataset with distribution shifts, as well as settings with low data regimes. Paper [3] which you cite presents the GTSRB dataset as one that has “minimal overlap with ImageNet-1K, with [. . .] 43 traffic signs aggregated into a single label in ImageNet-1K.” We ran our experiments on GTSRB and obtained the following results for 10 trials each, and our results are reported in the universal response.
>
> Additionally, we replicated the experiments in the paper [3] under low data settings. Following the experiments demonstrated by Figure 7 in [3], we randomly subsampled 10% and 50% of GTSRB’s training set and only used that subsample for training. Then, we tested on the test set. We ran these experiments for epsilon=0.1 and 0.7 and reported the results over 10 trials below.
>
> | Epsilon and Train Percentage Used | 0.1 at 10% | 0.1 at 50% | 0.7 at 10% | 0.7 at 50% |
> |:---------------------------------:|:----------:|:----------:|:----------:|:----------:|
> |                mean               |   28.33%   |   45.43%   |   37.98%   |   79.42%   |
> |                std                |  4.28E-03  |  1.42E-02  |  1.04E-02  |  6.48E-03  |
>
> As we can see when we compare our results to Figure 7 from [3] above (left hand side), in all four settings, we outperform on the test set for GTSRB.
>
> | Epsilon and Train Percentage Used |    0.1 at 10%    |     0.1 at 50%    |     0.7 at 10%    |     0.7 at 50%    |
> |:---------------------------------:|:----------------:|:-----------------:|:-----------------:|:-----------------:|
> |              our mean             |      28.33%      |       45.43%      |       37.98%      |       79.42%      |
> |           mean from [3]           | Approximately 5% | Approximately 20% | Approximately 27% | Approximately 56% |
>
> Q5: Not accounting for the privacy loss incurred in tuning the hyperparameters is a problem.
>
> A5: Thank you for your note. We noticed a lot of paper not accounting for the privacy budget of hyperparameters tuning, and due to limited computational resources, we did not include it.
>
> Q6: BLIP is definitely no more state-of-the-art as the authors claim.
> A6: Thank you for your comment. We will rephrase our language from “state-of-the-art” to “extension of CLIP.”  We acknowledge the feedback and want to re-emphasize that our focus in this paper is primarily to learn the private representations through contrastive loss instead of SOTA vision-language models such as LLaVa, which does not include contrastive loss.
>
> Q7: CLIP is relatively old. Novel variants of CLIP may yield zero-shot (epsilon = 0) performance that is higher and might reduce the room for improvements achievable under DP constraints. Could the authors consider the latest and strongest CLIP-inspired architectures?
>
> A7: While we acknowledge the evolution of CLIP-inspired architectures and the potential for newer variants to achieve higher zero-shot performance, our paper's primary focus lies in the integration of differential privacy  constraints into CLIP. Our intent is to explore the feasibility of learning private representations through contrastive loss rather than privately fine-tuning a pretrained model. In specific scenarios, such as health images, where the data is highly sensitive and substantially differs from publicly available datasets, even advanced CLIP variants may not exhibit better performance than the vanilla one. In such cases, there arises a need to privately train a model. Therefore, our choice of CLIP as a baseline aligns with our objective of addressing privacy concerns in specific contexts. We appreciate the suggestion and will certainly explore incorporating the latest architectures in future work while maintaining a keen focus on privacy-preserving methodologies.

---

> ### Author Response · Authors · 2023-11-23
>
> Q8: Since the gaussian mechanism is not applied at a sample level but at a mini-batch level, are the authors making a fair comparison between the epsilon values of their technique and the baselines?
>
> A8: Although the noise injection procedure is different, we ensure that the final output is $(\epsilon,\delta)$-DP by Proposition 3.1. This is through clipping the minibatch gradient. So we are in a fair comparison setting for privacy protection. However, we acknowledged that our experimental results rely on stronger pretrained models and additional information from text. Please also refer to our universal response with respect to the choice of baselines.
>
> Q9: Could the authors show the result of applying the contrastive clip fine-tuning for epsilon =∞? To know what's the upper bound on accuracy of the DP variant.
>
> A9: We have included the nonprivate results, and please refer to our universal response.
>
> Q10: Why the baselines selection varies based on the choice of the dataset?
>
> A10: Our intention is not to compare with these baseline methods, but rather to evaluate our proposed approach against the established performance standards on these datasets. Therefore, we reported some recently established performance standards on these datasets.
>
>
> [1] https://arxiv.org/pdf/2110.05679.pdf
>
> [2] https://arxiv.org/pdf/2212.06470.pdf
>
> [3] https://arxiv.org/abs/2306.03962
>
> [4] Lecuyer, M., Atlidakis, V., Geambasu, R., Hsu, D. and Jana, S., 2019, May. Certified robustness to adversarial examples with differential privacy. In 2019 IEEE symposium on security and privacy (SP) (pp. 656-672). IEEE.
>
> [5] Lu, Y., Huang, X., Dai, Y., Maharjan, S. and Zhang, Y., 2019. Blockchain and federated learning for privacy-preserved data sharing in industrial IoT. IEEE Transactions on Industrial Informatics, 16(6), pp.4177-4186.

---

### Official Review · Reviewer_WDu3 · 2023-11-01

**Soundness:** 3 good
**Presentation:** 3 good
**Contribution:** 3 good
**Rating:** 8
**Confidence:** 2

**Summary:**

The paper proposes a novel method called DP-CLIP to make CLIP training differentially private.
The method performs per-batch gradient clipping, which is needed as the CLIP loss is non-decomposable.
However, the implementation is reduced to setting the number of micro-batches to 1 in standard DP-SGD implementation, which is neat.
Empirically, the paper shows that DP-CLIP has a better privacy-utility trade-off than alternative methods in a setup based on pre-trained embedding followed by fine-tuning on private datasets, for classification and QA.
The paper also derives a theoretical bound for the privacy-utility under linear representation settings, which is novel as the loss function is not smooth as in other analysis of DP-SGD.

**Strengths:**

**significance** The paper studies an important problem of privacy in multi-modal learning with the widely-used CLIP loss and proposes a novel method to make the training DP. The proposed method shows promising privacy-utility trade-off empirically.

**originality** The proposed method and theoretical results in the paper are both novel. The bound derived in section 5 seems to be non-trivial given that the loss is not smooth.

**quality** The proposed method and the presented theoretical results are sound.

**clarity** The paper is well-written and easy to follow.

**Weaknesses:**

The comparison in table 2 seems to be unfair.
I assume the competitive methods does not use any pre-trained model so I'm not sure how to read the results.
Perhaps one should use the same pre-trained models and then apply the corresponding DP-method for fair comparison.

**Questions:**

I'm concerned that the private fine-tuning dataset is contained in the pre-training set, or at least very similar data.
Have the authors checked the following facts?
- Does the training set of the pre-trained CLIP contain MNIST-like images?
- How different is the vizwiz image captioning dataset from the training set?

---

> ### Author Response · Authors · 2023-11-23
>
> Q1: Does the training set of the pre-trained CLIP contain MNIST-like images?
>
> A1: Thanks for your comments! The original CLIP paper claimed that “CLIP was pretrained on a dataset of 400M image-text pairs, covering as broad a set of visual concepts as possible,” We realize that this means that CLIP's training set might be drawn from similar distributions to datasets such as MNIST, but we include zero-shot results above to show how much we can still privately learn through our presented training process. We have also included our results on GTSRB, a dataset with “minimal overlap with ImageNet-1K, with [. . .] 43 traffic signs aggregated into a single label in ImageNet-1K,” in the universal response above [1].
>
> Q2: How different is the vizwiz image captioning dataset from the training set?
>
> A2: Since the dataset used to train CLIP is not disclosed by OpenAI, we cannot answer how different the Vizwiz dataset is from the training dataset of CLIP. However, our additional experiments show that non-private fine-tuned scores are around 15% or above higher than zero-shot scores, suggesting that CLIP does not learn much from the Vizwiz dataset, or a similar dataset.
>
> [1] https://arxiv.org/abs/2306.03962

---

### Official Review · Reviewer_7tKv · 2023-11-04

**Soundness:** 3 good
**Presentation:** 3 good
**Contribution:** 4 excellent
**Rating:** 8
**Confidence:** 4

**Summary:**

This paper introduces DP-CLIP, a differentially private adaptation of the CLIP model for vision-and-language tasks, addressing privacy concerns while maintaining accuracy. CLIP and similar models have been shown to inadvertently disclose sensitive information, making privacy-preserving mechanisms crucial. DP-CLIP employs per-batch clipping to protect privacy and achieves strong performance on various benchmark datasets, such as image classification and visual question answering. The paper also discusses the privacy-utility trade-off under linear representation settings. Overall, DP-CLIP represents a important effort to enhance privacy protection in multimodal models, offering a significant reduction in data exposure risk while maintaining task performance.

**Strengths:**

1) The paper is the first to apply differential privacy approaches to multimodal training in the context of vision-language tasks, which is a much needed effort in enhancing privacy protection for such models.
2)  The paper conducts extensive experiments on different vision-language tasks, including image classification and image captioning, to evaluate the effectiveness of DP-CLIP across diverse datasets and privacy parameters. The paper provides a thorough comparison with related work in the field of differential privacy and vision-language tasks, ensuring that the contributions of DP-CLIP are well-placed
2) Paper is very well written and easy to follow.

**Weaknesses:**

1) The paper employs smaller classification datasets such as MNIST, Fashion-MNIST, CIFAR-10, and SVHN. It would have been preferable to observe results at the Imagenet scale, but I understand that the computational resources required for such experiments would have been substantial.

2) No comparison with real-world threat models has been provided. Epsilon-utility trade-offs can be misleading without testing them against actual attacks, as epsilon guarantees are built upon numerous assumptions, as indicated in [1].


Refs
-----
[1] A Critical Review on the Use (and Misuse) of Differential Privacy in Machine Learning (https://arxiv.org/abs/2206.04621)

**Questions:**

see weakness section

---

> ### Author Response · Authors · 2023-11-23
>
> Q1: The paper employs smaller classification datasets such as MNIST, Fashion-MNIST, CIFAR-10, and SVHN. It would have been preferable to observe results at the Imagenet scale, but I understand that the computational resources required for such experiments would have been substantial.
>
> A1: Thank you so much for your feedback! Upon your suggestion and suggestions from other reviewers, we additionally ran our methods on a larger dataset (CIFAR-100), as well as another dataset GTSRB.
>
> Q2: No comparison with real-world threat models has been provided. Epsilon-utility trade-offs can be misleading without testing them against actual attacks, as epsilon guarantees are built upon numerous assumptions, as indicated in [1].
>
> A2: We thank you for your comments. Previous work has suggested that DP can protect against common privacy attacks such as reconstruction attacks and membership inference attacks [2]. We agree that it would be nice to demonstrate empirical protection performance against attacks. However, most membership inference attacks rely on the shadow model technique, which requires training thousands of shadow models with the same architecture as the target model.  As it requires significant computing resources, we decide it is beyond the scope of this project, and we leave efficient attack techniques as future work.
>
>
> [1] A Critical Review on the Use (and Misuse) of Differential Privacy in Machine Learning (https://arxiv.org/abs/2206.04621)
> [2] Chen J, Wang WH, Shi X. Differential Privacy Protection Against Membership Inference Attack on Machine Learning for Genomic Data. Pac Symp Biocomput. 2021;26:26-37. PMID: 33691001

---

### Official Review · Reviewer_mwbE · 2023-11-07

**Soundness:** 1 poor
**Presentation:** 2 fair
**Contribution:** 1 poor
**Rating:** 1
**Confidence:** 4

**Summary:**

The authors proposed DP-CLIP, a method of continued pretraining CLIP in a privacy-preserving manner. They demonstrated the competitive performance of DP-CLIP on image classification and image captioning tasks, and proved the utility-privacy trade-off in a simplified learning setting.

**Strengths:**

- Studying the privacy risks in multimodal learning is a relevant and important research topic
- It is nice to a see a formal characterization of the utility-privacy trade-off

**Weaknesses:**

I found the experimental section to be very problematic. Specifically,
- Framing the algorithm as "DP-CLIP" is very misleading. You didn't train a CLIP model from scratch using DP-SGD, but rather continue pretraining a **well-trained** model in a privacy-preserving manner. This is not mentioned in the abstract, and is not made clear until the experimental section. As a side note, the comparison with Yu et al. (2023) is unfair: 1) there are obviously no "two stages" in your algorithm since you only performed continued pretraining; 2) Yu et al. (2023) only pretrained on synthesized textures to speedup the DP training process, while the starting point of your DP training is already a very strong model.
- The classification tasks are too simple given the power of the pretrained model. It might not be necessary at all to perform private continued pretraining; plus I found pretraining on datasets such as MNIST and CIFAR-10 to be super weird. Two critical baselines are currently missing: the accuracy of the vanilla CLIP model (without continued pretraining), and the accuracy of $\varepsilon=\infty$ (non-private continued pretraining).
- The comparison with other baselines in Section 4.2 is arbitrary and careless. The authors didn't evaluate the same set of methods on all datasets, and didn't report whether the results for other algorithms are based on their own implementation or directly copied from prior works. The accuracy of DP-Sinkhorn on Fashion-MNIST does not match the one reported in the original paper. It is also unclear whether these algorithms are indeed the state-of-the-art methods in image classification. Finally, DP-Sinkhorn is not even a DP image classification method -- while it is capable of performing such task, it is essentially an algorithm of DP data synthesis (take a look at [1] if you are not familiar with this concept). It is unbelievable that the authors chose this method for comparison.
- I didn't buy the results from Section 4.3; particularly, I have never seen using $\varepsilon = 1e-4$ in the privacy literature, and it is hard to believe that such privacy budget could still lead to a meaningful model (in the extreme case $\varepsilon=0$, the model will become completely random). On the other hand, the authors seemed to suggest that the model's performance is mostly unaffected even in this extreme case. I insist: 1) report the $\sigma$ (the noise multiplier), and check the SNR to see whether the results make sense; 2) use $\varepsilon=1e-8$ and see whether the results are still consistent; 3) use the vanilla CLIP model (without any further continued pre-training) and see whether the results are already very good. Other issues in this section: 1) there is no description of the non-private model, I have completely no idea what "IBM Research AI" is and whether the comparison with DP-BLIP makes sense at all; 2) it is unclear why a baseline of non-private BLIP is missing.

Minor:
- The experiments should be running over multiple random seeds, and please include the standard deviations in the tables
- The dataset paragraph should be revised. The descriptions of the datasets are way too long yet important details are missing (e.g., the dimensionality of each dataset). The metrics used in image captioning are not explained.
- "TensorFlow Privacy": which privacy accountant did you use exactly? As a side note, RDP and PRV accountant are the go-to choices nowadays.
- The paragraph below Table 2: "which is not present for DP-SGD" -- please note that the pretraining and extra caption data are not used in other baseline methods as well
- Various grammatical issues, included but not limited to: "while a smaller or equal $\varepsilon$" -> "with a smaller or equal $\varepsilon$"; "allow our private approach to achieving" -> "to achieve"; "Apart from their work" -- I don't understand what you meant here; "requires... should be indistinguishable" --> "to be indistinguishable". Please do a professional proofreading before submitting your work.

In light of the major issues in the experimental section, this paper is clearly below the bar of ICLR and I will strive for a rejection.

Reference

[1] Hu et al. "SoK: Privacy-Preserving Data Synthesis." 2024 IEEE Symposium on Security and Privacy (SP). IEEE Computer Society, 2023.

**Questions:**

See above

---

> ### Author Response · Authors · 2023-11-23
>
> Q1: Framing the algorithm as "DP-CLIP" is very misleading.
>
> A1: We would like to emphasize that our proposed algorithm and the associated theory are designed for differentially private CLIP training from scratch. In the experimental section, we opted to use a pretrained model as an initialization solely to reduce computational costs. It's important to note that our intention was not to compare our method against Yu et al. (2023) due to the inherent differences in the addressed problem. Our mention of their work is specifically to highlight that, like ours, their framework also involves using a pretrained model as initialization.
>
> Q2: The classification tasks are too simple given the power of the pretrained model.
>
> A2: We have included both the zero-shot results and the non-private results. we also added two new datasets: Cifar 100 and GTSRB, where the GTSRB dataset presents a significant distribution shift with respect to the pre-training set. Please refer to our universal response.
>
> Q3: The comparison with other baselines in Section 4.2 is arbitrary and careless.
>
> A3: Thank you for pointing it out. We found several versions of the DP-Sinkhorn paper online: 1) from neurips https://proceedings.neurips.cc/paper_files/paper/2021/file/67ed94744426295f96268f4ac1881b46-Paper.pdf and 2) from arxiv https://arxiv.org/pdf/2111.01177.pdf. We used results from the newer version that appeared at NeurIPS in our paper. Looking at Table 1 from neurips version, we got an accuracy of 83.2 on MNIST and 73.0 on Fashion MNIST for epsilon=10 as cited in our paper (the numbers were taken from the best results with different architectures). If we look at the arxiv version, we get 83.2 on MNIST and 74.6 on Fashion MNIST for epsilon=10. Regarding our choice of the baseline methods, please refer to our universal response.
>
> Q4: I didn't buy the results from Section 4.3.
>
> A4: Thank you for your suggestions. We attach a set of more thorough experiments, including zero-shot results, eps = 1e-4, 1, 5, and non-private results. Please refer to the table in the universal response. Our additional experimental findings align with the results presented in the paper, demonstrating that post-training consistently outperforms zero-shot performance for epsilon values ranging from 1e-4 to $\infty$ (non-private.) A consistent trend is observable across various metrics. However, we did implement the 1e-8 regime for multiple trials and found that the result was pretty random, which suggests that the noise is substantial enough to compromise the model's utility. As a side note, due to the limit of time and computational resources, this set of additional results was generated using a training set of 5K samples and a test set of 1K samples, which results in a performance gap compared to our original results.
>
> Regarding the state-of-the-art non-private performance provided by IBM Research AI, it is a score from the Kaggle website (https://paperswithcode.com/sota/image-captioning-on-vizwiz-2020-test). It is the only comparable baseline we could find online, although it is unclear what their model is. We will add this link to the manuscript.
>
> Q5: The experiments should be running over multiple random seeds.
>
> A5: Thank you for your suggestion — we have re-run all our experiments using 10 trials and report the mean and standard deviations in the universal response.
>
> Q6: The dataset paragraph should be revised.
>
> A6: We thank you for your suggestions. We will revise the description for the datasets and add explanations on evaluation metrics for the image captioning task.
>
> Q7: Which privacy accountant did you use exactly?
>
> A7: We are using RDP accountant implemented at tensorflow privacy package (https://github.com/tensorflow/privacy/blob/master/tensorflow_privacy/privacy/analysis/compute_noise_from_budget_lib.py .)
>
> Q8: The paragraph below Table 2: "which is not present for DP-SGD" -- please note that the pretraining and extra caption data are not used in other baseline methods as well.
>
> A8: We acknowledge that there are no “fair” baselines to compare with. Our intention is not to compare with their methods, but rather to evaluate our proposed approach against the established performance standards. We will further clarify this in this paragraph.
>
> Q9: Various grammatical issues
>
> A9: We have fixed these grammar issues and did a thorough check on the full paper.

---

### Author Response · Authors · 2023-11-23

# Universal Response:

We thank all reviewers for their valuable comments and suggestions. We would like to start by addressing some common misunderstandings about our paper that were mentioned across reviewers. For specific details, please refer to the individual responses.

We added sets of new experimental results for image classification and image captioning. For image classification, we added zero-shot and non-private results on the current datasets, and also re-run all our experiments using 10 trials, and reported the mean and standard deviations for private results as we varied the privacy parameters. Following Reviewer 68X7’s suggestions, we also added two new datasets: Cifar 100 and GTSRB, where the GTSRB dataset presents a significant distribution shift with respect to the pre-training set [1]. For image captioning, we included zero-shot results, epsilon = 1e-4, 1, 5, zero-shot and non-private results. These results are listed below. From the table below, we can observe the trade-off between privacy and utility, consistent with what we showed in the paper.

Regarding the novelty in our paper, we would like to highlight that to the best of our knowledge, our paper is the first to theoretically establish the (optimal) privacy-accuracy trade-off for CLIP training while maintaining DP, supported by our theory and experiments. We acknowledge that other benchmarks in our paper are not proposed for classification tasks, like ours. We emphasize that our proposed method uniquely addresses the DP training of contrastive vision-language models, not only limited to DP image classifications. From this point, there are no “fair” baselines to compare with. We specifically chose these benchmarks because they demonstrate State-of-the-Art results in image classification tasks on these datasets. Our intention is not to compare with their methods, but rather to evaluate our proposed approach against the established performance standards.

[1]  https://arxiv.org/abs/2306.03962

# Image Classification Results:
### Baselines:
| epsilon |   MNIST   |    Fashion-MNIST   |     Cifar 10    |     SVHN    |
|:-------:|:--------:|:--------:|:--------:|:--------:|
|   zero-shot  |  48.06%  |  66.31%  |  88.31%  |  24.86%  |
|   non-private   |  98.97% |  92.08% |  95.07% |  93.81% |


### MNIST:
| epsilon |   0.25   |    0.5   |     1    |     3    |    10    |
|:-------:|:--------:|:--------:|:--------:|:--------:|:--------:|
|   Mean  |  71.55%  |  74.62%  |  76.29%  |  78.22%  |  79.68%  |
|   Std   | 3.16E-03 | 3.05E-03 | 3.37E-03 | 2.57E-03 | 2.46E-03 |


### Fashion-MNIST:
| epsilon |   0.25   |    0.5   |     1    |     3    |    10    |
|:-------:|:--------:|:--------:|:--------:|:--------:|:--------:|
|   Mean  |  88.83%  |  89.76%  |  90.35%  |  91.05%  |  91.42%  |
|   Std   | 1.35E-03 | 2.28E-03 | 1.44E-03 | 1.76E-03 | 1.56E-03 |

### Cifar 10:
| epsilon |   0.25   |    0.5   |     1    |     3    |    10    |
|:-------:|:--------:|:--------:|:--------:|:--------:|:--------:|
|   Mean  |  93.19%  |  94.03%  |  94.74%  |  95.16%  |  95.48%  |
|   Std   | 3.67E-03 | 1.36E-03 | 1.19E-03 | 1.20E-03 | 1.18E-03 |

### SVHN:
| epsilon |   0.25   |    0.5   |     1    |     3    |    10    |
|:-------:|:--------:|:--------:|:--------:|:--------:|:--------:|
|   Mean  |  87.18%  |  88.92%  |  90.31%  |  91.56%  |  92.53%  |
|   Std   | 2.93E-03 | 3.14E-03 | 2.59E-03 | 1.79E-03 | 1.16E-03 |

### Cifar 100:
| epsilon |   0.25   |    0.5   |     1    |     3    |    10    |
|:-------:|:--------:|:--------:|:--------:|:--------:|:--------:|
|   Mean  |  71.55%  |  74.62%  |  76.29%  |  78.22%  |  79.68%  |
|   Std   | 3.16E-03 | 3.05E-03 | 3.37E-03 | 2.57E-03 | 2.46E-03 |

### GTSRB:
| epsilon |   0.25   |    0.5   |     1    |     3    |    10    |
|:-------:|:--------:|:--------:|:--------:|:--------:|:--------:|
|   Mean  |  77.55%  |  84.95%  |  88.59%  |  91.31%  |  92.47%  |
|   Std   | 7.64E-03 | 6.93E-03 | 2.85E-03 | 5.90E-03 | 2.50E-03 |

# Image Captioning Results
| epsilon | zero-shot |  1e-4 |   1   |   5   | non-private |
|:-------:|:---------:|:-----:|:-----:|:-----:|:-----------:|
|  Bleu_1 |   0.587   | 0.659 | 0.665 | 0.668 |    0.679    |
|  Bleu_2 |   0.406   | 0.475 | 0.478 | 0.484 |    0.492    |
|  Bleu_3 |   0.270   | 0.333 | 0.333 | 0.338 |    0.343    |
|  Bleu_4 |   0.171   | 0.222 | 0.222 | 0.230 |    0.231    |
| ROUGE_L |   0.417   | 0.454 | 0.457 | 0.475 |    0.463    |
|  CIDEr  |   0.431   | 0.584 | 0.584 | 0.690 |    0.695    |
|  SPICE  |   0.165   | 0.189 | 0.193 | 0.210 |    0.207    |

---

### Meta-Review · Area_Chair_oBEo · 2023-12-12

**Metareview:**

(a) Summarize the scientific claims and findings of the paper based on your own reading and characterizations from the reviewers.

The authors proposed DP-CLIP, a method of continued pretraining CLIP in a privacy-preserving manner. They demonstrated the competitive performance of DP-CLIP on image classification and image captioning tasks, and proved the utility-privacy trade-off in a simplified learning setting.

(b) What are the strengths of the paper?

This paper is the first to make contrastive learning private, and showcase empirical performance comparable or better than other methods.

(c) What are the weaknesses of the paper? What might be missing in the submission?

Some comparisons are not fair given that the proposed approach starts from a pretrained CLIP models. The experiments are on finetuning, but for some reason not as a classifier but with CLIP training. Given this, some important baselines for finetuning CLIP are not provided. The evaluations of the proposed method leaves room for improvement. Some important comparisons are not reported. Per batch clipping is obviously lossy, since individual gradient's contribution, typically, should not match the worstcase batch sensitivity. Proposition 3.1 trivially follows.

**Justification For Why Not Higher Score:**

The proposed method clips minibatches and add appropriate noise in private contrastive learning. This part is straightforward. The novelty is in the reported performance gain of such approach. However, the paper fails to deliver the promised gain. First of all, CLIP training is used to finetune on small datasetes a pretrained CLIP model. This goes against the motivation of private contrastive training. Some obvious baselines are missing, given we are comparing finetuning tasks. This makes it hard to discern where the gain is coming from.

**Justification For Why Not Lower Score:**

N/A

---

### Decision · Program_Chairs · 2024-01-16

Reject